Analysis

# Causal inference concepts can guide research into the effects of climate on infectious diseases

**Laura Andrea Barrero Guevara[1,2], Sarah C. Kramer [1], Tobias Kurth [2] & Matthieu Domenech de Cellès [1]** ✉

A pressing question resulting from global warming is how climate change will affect infectious diseases. Answering this question requires research into the effects of weather on the population dynamics of transmission and infection; elucidating these effects, however, has proved difficult due to the challenges of assessing causality from the predominantly observational data available in epidemiological research. Here we show how concepts from causal inference—the sub-field of statistics aiming at inferring causality from data—can guide that research. Through a series of case studies, we illustrate how such concepts can help assess study design and strategically choose a study's location, evaluate and reduce the risk of bias, and interpret the multifaceted effects of meteorological variables on transmission. More broadly, we argue that interdisciplinary approaches based on explicit causal frameworks are crucial for reliably estimating the effect of weather and accurately predicting the consequences of climate change.

A key question ensuing from global warming is how climate change may impact the population dynamics of infectious diseases[1–3]. Indeed, observations of large climatic variability in the distribution and seasonality of multiple infectious diseases worldwide—including major causes of death like malaria[4], cholera[5] and influenza[6]—suggest that many pathogens are sensitive to environmental conditions such that climate change could modify their ecology and epidemiology. Accordingly, predictive studies, based on numerical simulations combining models of global climate and infectious diseases under different scenarios of greenhouse gas emissions, suggest that climate change will affect many infections. These include infections with indirect transmission through intermediate, climate-sensitive stages involving a vector, such as mosquito-borne diseases like malaria and dengue, or the environment, including water-borne diseases like cholera. All of these infections are predicted to shift their geographical range under continued global warming[7–9]. Fewer studies have focused on directly transmitted pathogens, but it has been suggested that climate change could also alter the transmission dynamics of respiratory syncytial viruses in the United States and Mexico[10] and varicella zoster viruses in Mexico[11]. Although such predictions cannot yet be evaluated, earlier research has already documented the impact of past climate warming[3], for example, the increased altitudinal range of malaria in the highlands of Ethiopia and Colombia[12] and the increased risk of *Vibrio* disease in Northern Europe, coinciding with the warming of the Baltic Sea's surface[13].

A prerequisite to predicting the long-term consequences of climate change is to elucidate the effect of weather on infectious diseases. Even though effects of weather on infection dynamics (and the resulting 'calendar of epidemics'[14]) have long been observed, there are persisting uncertainties about the direct causes and mechanisms for even well-researched pathogens like influenza viruses[15,16]. Perhaps the most robust evidence for this effect is afforded by experimental studies, which demonstrate that environmental variables like temperature and humidity tightly modulate transmission parameters (such as pathogen survival time or infectivity) of viral[17,18], bacterial[19,20] and parasitic[21,22] infections. Although such evidence is useful for population-based

[1]Max Planck Institute for Infection Biology, Infectious Disease Epidemiology Group, Campus Charité Mitte, Berlin, Germany. [2]Institute of Public Health, Charité–Universitätsmedizin Berlin, Berlin, Germany. ✉e-mail: domenech@mpiib-berlin.mpg.de

research (in particular for postulating causal environmental factors), it remains too limited to estimate the causal impact of weather at the scale of human populations for at least three reasons. First, the end-points measured in experimental studies—such as pathogen survival time—can be challenging to translate into meaningful epidemiological quantities, such as transmissibility. Second, because of differences in infection biology between species, the results from animal studies may not generalize to humans[23]. Third, experimental studies cannot recapitulate all the mechanisms whereby weather affects infection, especially those operating at the population level—for example, weather causes behavioural changes in people, resulting in seasonal changes in social contacts[24]. Hence, observational studies remain necessary to estimate the multifaceted effects of weather on human infectious diseases.

However, a well-known shortcoming of observational studies is their tendency to misidentify causes because found associations do not always imply causation for observational data. This problem is also expected when inferring the effect of weather, which is characterized by meteorological variables generally highly correlated with one another and potentially many other seasonal causes of infectious diseases. Here we discuss and demonstrate how causal inference—a methodological framework aiming at inferring causes from observed data—offers a principled approach to tackle these issues and strengthen evidence in observational research[25,26].

## Causal inference in climate–infectious disease research

Causal inference frameworks and their tools are increasingly used to analyse data and guide study design in epidemiology[27] and beyond[28,29], and may also be useful for experimental research[29]. The impact of such tools is illustrated by the fact that causal frameworks like target emulation trials may provide evidence as robust as that from randomized trials[30], thereby expanding the scope of observational research for answering causal questions.

Despite these advances, the use of causal methods—in the form of mechanistic models or statistical models based on causal reasoning—remains limited in the field of weather- or climate–infectious disease research[31]. To assess how regularly causal methods are used in these fields, we re-analysed 33 studies previously assessed in a review[32–64]. All of these studies used time series regression models to evaluate the association between weather and cases of dengue, influenza, cholera or malaria cases (Supplementary Table 1). Although more causally principled methods for time series analysis exist[65,66], the standard time series design remains widely used, as evidenced by numerous recent applications for SARS-CoV-2[67]. Four of the 33 studies had an explicitly predictive objective and did not address causality[38,41,43,58]. Of the remaining 29 studies that addressed causality, only one derived the statistical model from explicit causal assumptions[40]. By contrast, the other 28 studies neither explicitly mentioned causal reasoning to formulate their research question nor used causal graphs for study design or statistical analysis. Based on this assessment, we conclude that applying causal inference methods may help strengthen the evidence in this field.

Here we aim to demonstrate how using a causal inference framework can improve research on the effects of weather and climate on infectious diseases. Our focus is on the conceptual aspects, while we recommend other studies for methodological details[65,68,69]. Throughout our analysis, we illustrate the different causal inference concepts through a series of short case studies ('vignettes'), all based on a causal inference framework described below.

## Results

### A causal inference framework for a model of infectious disease transmission

A causal inference framework can be broadly defined as a systematic approach to identifying the causal effects of an input variable (for example, a weather variable) on an outcome variable of interest (for example, infectious disease risk)[70,71]. In this framework, a central first step is to explicitly represent cause-and-effect relationships using a causal diagram called a directed acyclic graph (DAG; see glossary Fig. 1)[72]. The benefits of this representation are double: conceptual, as they help investigators clarify their hypotheses and assumptions and highlight potential sources of bias; and methodological, as given a DAG, causal inference theory prescribes a set of rules to determine the form of a model (if any) that can answer the causal question of interest. In a DAG, one should also carefully consider the measurement process to distinguish between observed and unobserved variables. This distinction is central for infectious diseases, as variables that are typically unobserved—such as population immunity, often portrayed as the 'dark matter' of epidemics[73]—sensitively control transmission dynamics. We illustrate the application of a causal inference framework with a simple mathematical model representing the population-level dynamics of an acute infection spread by direct contact between susceptible and infected hosts. Although we do not focus on a specific pathogen in the rest of this analysis, this model—known as Susceptible–Infected–Recovered–Susceptible (SIRS) in the field of infectious disease modelling[74]—may be considered realistic for respiratory viruses with short generation times, like influenza viruses[75], respiratory syncytial viruses[76] and SARS-CoV-2[76].

To describe the effect of weather on infection dynamics, we incorporated an environmental model representing the joint causal effect—dictated by physical laws[77,78]—of ambient air temperature (Te) and dew point temperature (a measure of absolute humidity) on relative humidity (RH). We then assumed a direct, negative effect of temperature and relative humidity on transmission ($\beta$)—that is, transmission decreased as either climatic variable increased. Finally, we incorporated an observation model representing the causal link between the true and observed incidence rates, assuming a surveillance system with perfectly specific but incompletely sensitive case detection, resulting in case under-reporting. Of note, we assumed this observation model was completely random, and we thus did not consider the potential bias resulting from systematic differences between the true and the observed incidence rates[79–81]. Strategic simplifying assumptions (in particular, of a discrete-time model with fixed generation time and time step of 1 week) then allowed us to represent the full model with a simple DAG (Fig. 2). We focus on this representation instead of the conventional compartmental model diagram used in infectious disease modelling because DAGs more concisely convey causal concepts. Still, we note that both representations are causal diagrams (see ref. 82 for a more extensive discussion of the correspondence between both diagrams). Full model details—including equations, numerical implementation and further discussion of our assumptions about the effects of weather—can be found in Methods.

In the rest of this analysis, we use this DAG and its underlying model to illustrate four causal inference concepts: descendants and measurement bias (vignette 1); natural experiments (vignette 2); confounders and confounding bias (vignette 3); and mediators (vignette 4). In so doing, we emphasize two key points: first, causal inference frameworks are useful—and, indeed, required—to assess the effects of weather on infectious diseases; second, transmission models are valuable to encapsulate this framework, as they specify explicit causal mechanisms for the observed and unobserved variables that underlie infectious disease data.

### Causal inference concepts–illustrations with four vignettes
**Vignette 1 on descendants, measurement bias and the intricate association between environmental variables and incidence rate.**
Time series regression analysis of observed incidence rates is a frequent study design in environmental epidemiology[68,83]. The implicit assumption of such studies is that statistical quantities derived from regression models—typically, regression coefficients—will accurately

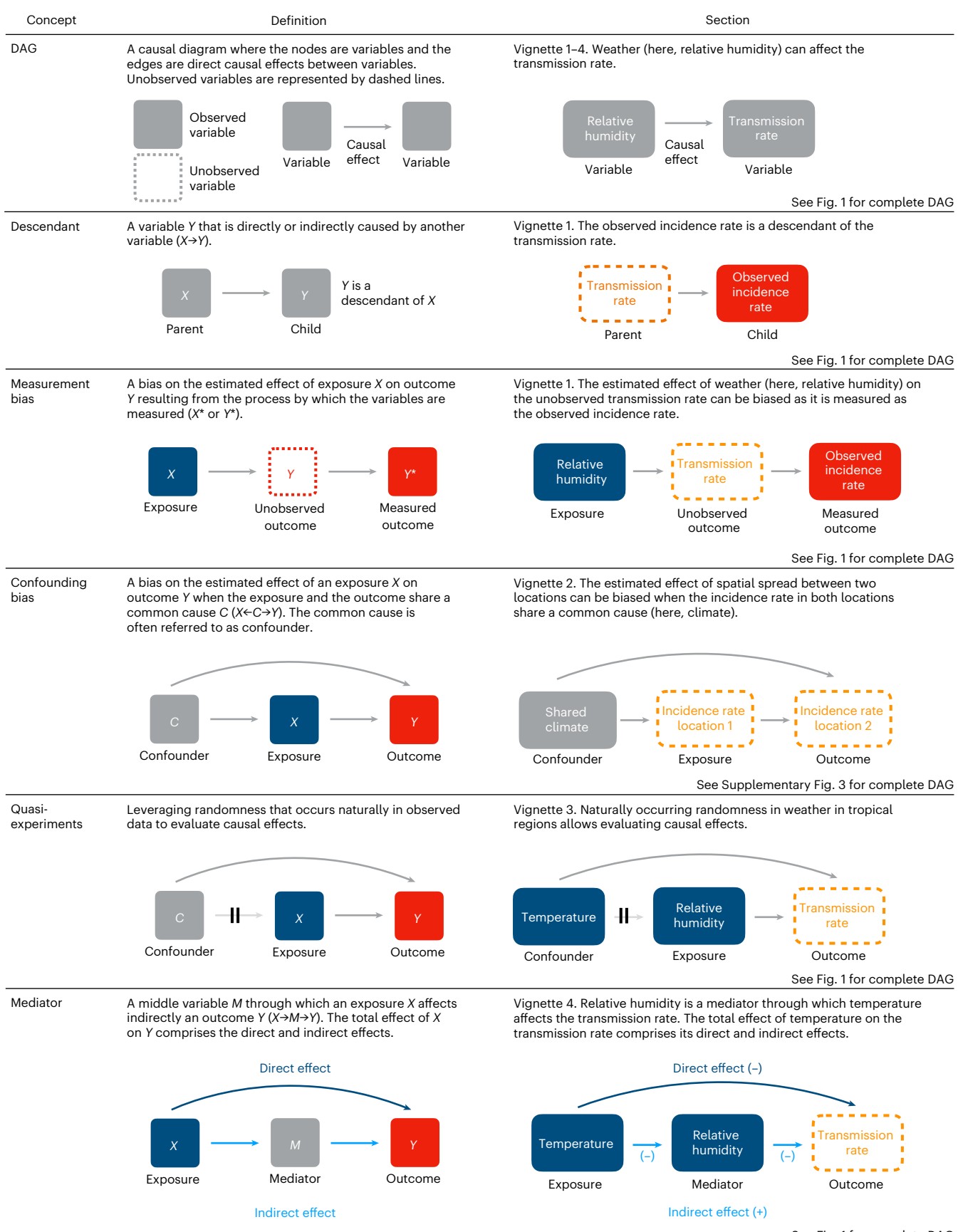

**Fig. 1 | Glossary of causal inference concepts.** Definitions of causal inference concepts with causal graphs illustrating the topic discussed in each vignette.

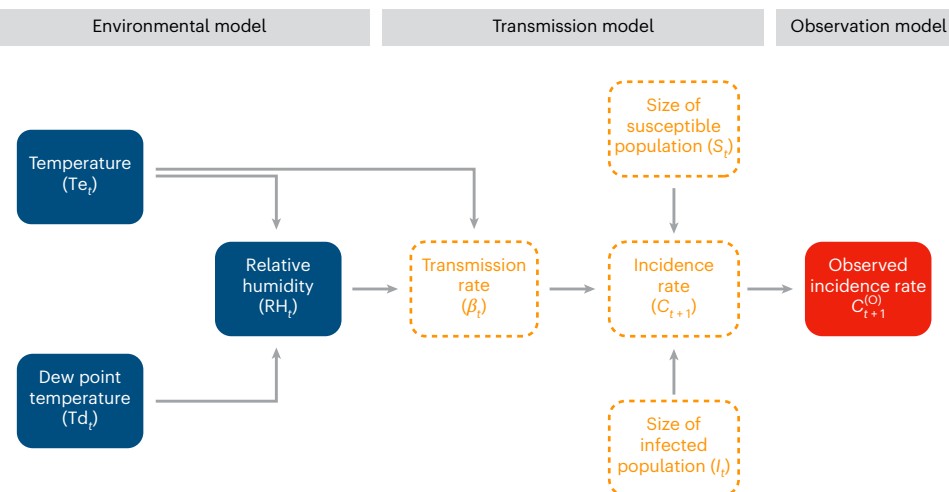

**Fig. 2 | Causal graph for the illustrative transmission model.** At a given time $t$, physical laws[77,78] dictate that the ambient temperature and the dew point temperature jointly determine the relative humidity of the air. We assume that both temperature and relative humidity directly affect the transmission rate of the pathogen: $\log \beta_t = \log \beta + \delta_{Te}(Te'_t - 1) + \delta_{RH}(RH'_t - 1)$, where $\beta$ represents the average transmission rate, $Te'_t$ and $RH'_t$ the rescaled environmental variables (with unit mean), and $\delta_{Te} = \delta_{RH} = -0.2$ their causal effects on transmission. The causal link between the transmission rate and the incidence rate—defined as the number of new cases per unit of time—is governed by a deterministic nonlinear model (here assumed to be a simple SIRS model[74]) representing the pathogen's epidemiological dynamics, in which new infections arise from contact between susceptible and infected individuals. Because of observation error (for example, imperfect test sensitivity resulting in case under-reporting), the observed incidence rate differs from the actual incidence rate; a stochastic observation model (here assumed to be a negative binomial model) represents the causal link between the two rates. Mathematically, this causal diagram translates into a discrete-time model that iterates the epidemic dynamic from one generation of infection to the next (see Methods for full information about the model's formulation and implementation). Variables surrounded by dashed lines are assumed to be unobserved.

capture the causal effect of meteorological variables. However, as shown in our causal graph (Fig. 2) and as expected in practice, the causal chain relating weather to observed incidence rates is indirect and complex. A key issue arises from the fact that, although weather directly affects the transmission rate, the observed incidence rate is located two causal links downstream from the transmission rate; in causal inference language, the latter rate is described as a descendant (that is, a consequence; Fig. 1) of the former. Of these two causal links, the one relating the transmission and incidence rates may be challenging to recapitulate with regression models because unobserved variables like the size of the population susceptible to infection (inversely related to the population, or herd, immunity, which controls epidemic thresholds) induce nonlinearities that may result in marked dissimilarities between these two rates[69,74]. Hence, measurement bias—that is, the bias arising from non-random differences between the targeted (here, transmission) and the observed endpoints (here, observed incidence; Fig. 1)[84]—may distort causal inference from time series regression models. This potential bias has been recognized in environmental epidemiology, as reflected in recommendations to include additional covariates for capturing temporal variations in population immunity or other long-term trends[68]. However, because of the above complexities, such additions, depending on the underlying causal structure and available information, are not guaranteed to reduce measurement bias.

To illustrate, we generated model simulations for a pathogen with low, medium or high transmissibility (basic reproduction number of 1.25, 2.5 or 5, respectively), with meteorological data from a temperate climate (Lübeck, Germany; Supplementary Table 2) resulting in a seasonally forced transmission rate with a single peak every winter (Fig. 3a). Under the medium-transmissibility scenario (Fig. 3b, middle panel), the epidemiological dynamic displayed annual periodicity, with winter seasonality in the incidence rate that broadly matched that of the transmission rate (Spearman's correlation coefficient: $r_s = 0.63$). In marked contrast, lower transmissibility resulted in biennial epidemics showing little correlation with the seasonal transmission rate ($r_s = 0.07$; Fig. 3b, top panel). This phenomenon—called sub-harmonic resonance[74]—resulted from the higher susceptibility threshold needed

to trigger epidemics and the longer time required to replenish the pool of susceptible individuals (via births and waning immunity) to exceed that threshold. Finally, the opposite phenomenon of super-harmonic resonance was observed in the high-transmissibility scenario, which resulted in biannual epidemics (Fig. 3b, bottom panel). These simple numerical experiments illustrate the complex dynamic of infectious diseases and the potent but sometimes counter-intuitive footprint that weather—or, for that matter, any other source of seasonal forcing—can have on this dynamic[69].

Next, we generated 100 replicate time series of observed incidence rates for each scenario to assess the reliability of time series regression models. For every replicate, as a control, we first fitted a negative binomial generalized additive model (GAM) with 1-week-lagged weather variables and the susceptible and infected population sizes as covariates and the observed incidence rate as endpoint—the true candidate model for our application, as we show in the Methods. As expected, the causal effects of temperature and relative humidity on the transmission rate were estimated, on average, without bias for this model (Supplementary Fig. 1). In practical applications, however, the susceptible and infected population sizes would be unobserved. Therefore, we next fitted a comparable model with a flexible smooth of time to try to capture variations in these unobserved variables. As shown in Fig. 3c, because of measurement bias, estimation performance was, overall, poor. For low transmissibility, the causal effect of temperature on the transmission rate was estimated with a substantial bias (mean absolute bias (AB): 0.08, 40% relative error in comparison to the actual value of −0.2) and imprecision (mean standard error of estimates across simulations (SE): 0.07). The bias was even more substantial in the medium- (mean AB: 0.16) and high-transmissibility scenarios (mean AB: 0.22). Because relative humidity had a high correlation with temperature but lower variability, its estimated effect was marred with large uncertainty, which exceeded, on average, the absolute effect size in every scenario (mean SE: 0.21–0.35, to be compared with the true effect size of −0.2; Supplementary Fig. 1).

In an additional analysis, we tested time series regression models of the effective reproduction number, another outcome that can be

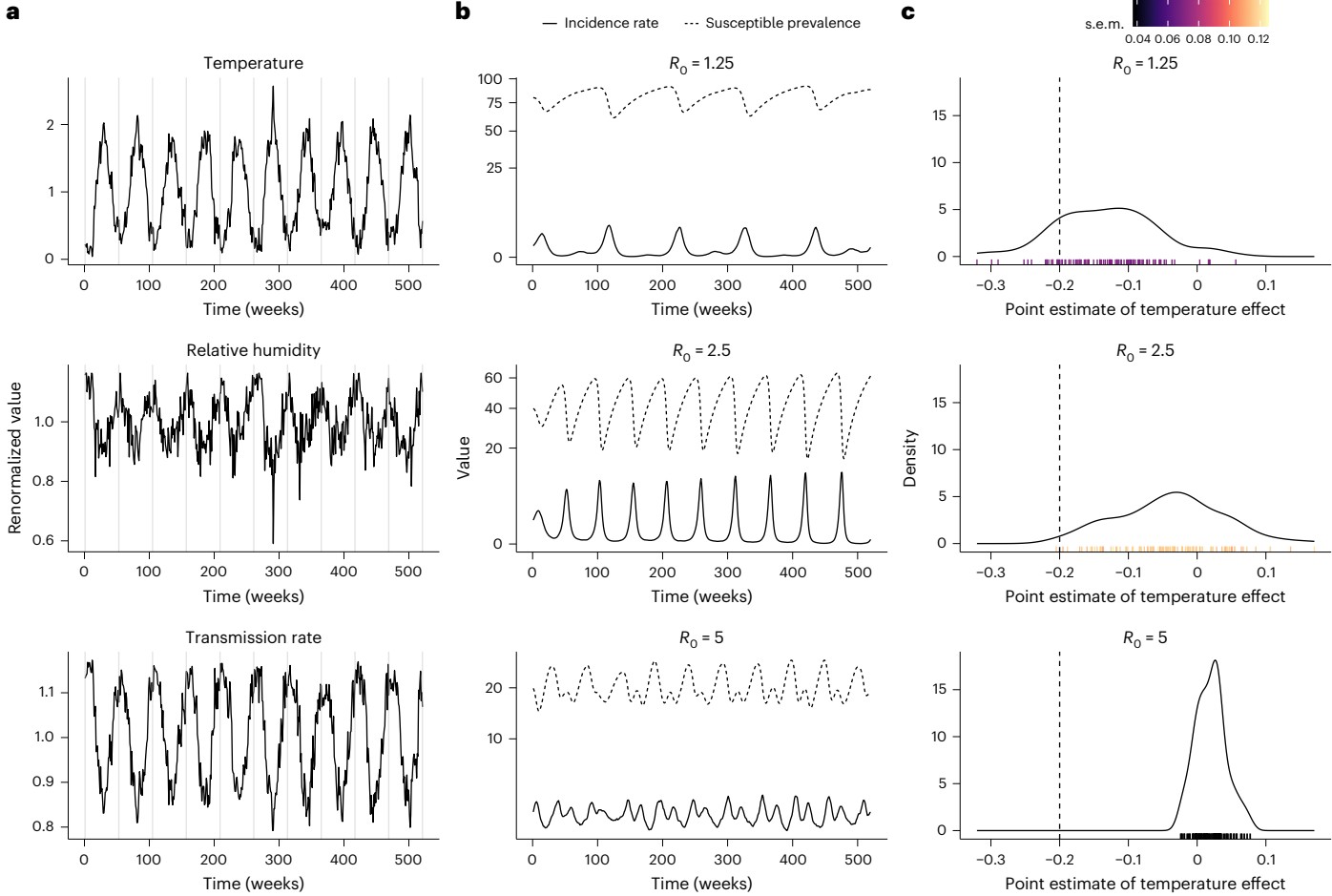

**Fig. 3 | Measurement bias and the intricate association between environmental variables and incidence rate (vignette 1). a**, Time series of temperature (top panel) and relative humidity (middle panel, both climatic variables renormalized to have unit mean) during 2013–2022 in Lübeck, Germany. The bottom panel displays the resulting seasonal component of the transmission rate, assuming a negative effect of both climatic variables on transmission. In the three panels, the vertical grey lines mark the beginning of every calendar year (week no. 1). **b**, Dynamics of susceptible prevalence ($100 \times S_t/N$; dashed lines) and incidence rate ($100 \times C_t/N$; solid lines) for three different values of the basic reproduction number ($R_0$) and an average duration of immunity of 1 year (other model parameters fixed to the values in Supplementary Table 3). **c**, Distribution of point estimates for the effect of temperature estimated from a negative binomial GAM regression model fitted to each of 100 replicate time series of the observed incidence rate. The marks on the $x$ axis indicate the point estimates, with the colours representing their standard errors. The vertical dashed line indicates the true effect fixed in the transmission model for generating the observations.

considered to assess the effects of weather. The corresponding DAG (Supplementary Fig. 2) shows that this outcome depends on only one unobserved variable, while the incidence rate depends on two (Fig. 2). As a result, we found that time series regression generally performed better for this outcome, even though the bias remained very large in all scenarios (Supplementary Fig. 1). We note that this better performance may not generalize to all settings. In practice, the effective reproduction number is not directly observable and must first be reconstructed from incidence time series[85]. In our deliberately simple application, we assumed a small amount of noise but no systematic bias in this reconstruction, an assumption that may be too optimistic for more realistic models.

Although not intended to be exhaustive, this simple simulation study echoes earlier discussions of measurement bias as a major concern for study designs based on time series regression, particularly when population immunity varies over fast time scales[68,83]. Relating to the central thread of this analysis, we note that causal reasoning—particularly the causal diagram representing the effect of weather—allowed us to identify, a priori, the relevant theoretical issues and propose simple numerical experiments to assess their practical relevance. This vignette thus illustrates the value of causal reasoning not only as a methodological tool but also as a theoretical tool to guide study design and analysis.

**Vignette 2 on climate variability as natural experiments to estimate the individual effect of meteorological variables.** Owing predominantly to latitudinal gradients in solar radiation and other factors like altitude and proximity to the sea, the Earth displays a large variability of climates[86]. This variability is reflected in the Köppen–Geiger system, which classifies worldwide climates into 5 main types and 30 sub-types based on seasonal averages of precipitation and temperature[87]. Because these different climates exhibit diverse seasonal patterns of variation in weather variables and correlations between them, they may be regarded as a range of 'natural experiments,' conceptually equivalent to manipulating specific weather variables to identify their causal effects. More broadly, the strategy of leveraging randomness that occurs naturally in observed data (that is, quasi-experiments; Fig. 1) is increasingly advocated for when inferring causality in predominantly observational research fields like economics[28], ecology[88] and epidemiology[89]. As an example of such quasi-experiments, previous studies analysed large-scale, irregular oceanic phenomena such as the El Niño–Southern Oscillation to evaluate the effects of 'climate change-like shocks'[90,91].

Of particular interest for environmental epidemiological research is the contrast between tropical climates (where temperature generally varies little and thus only slightly affects relative humidity) and temperate climates (where the opposite is typically observed), which may be

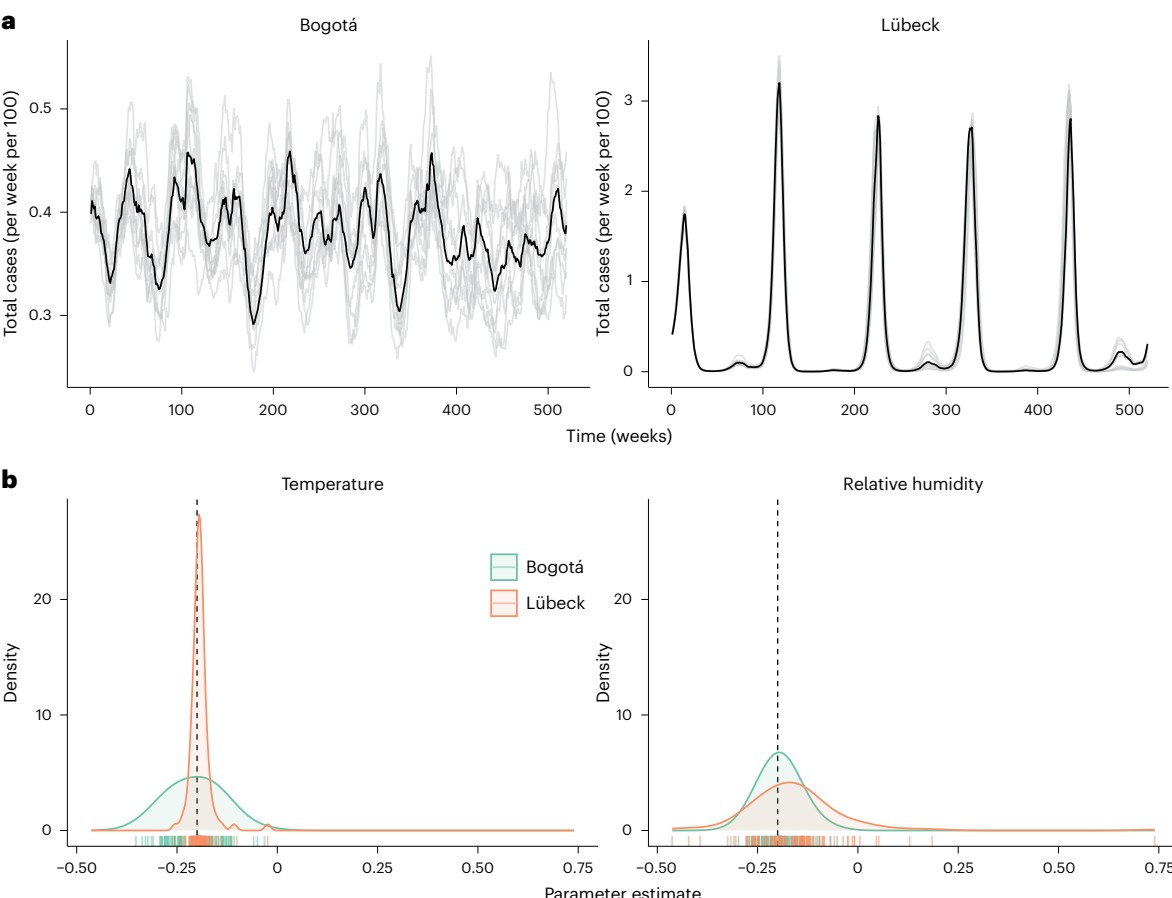

**Fig. 4 | Climate variability as natural experiments to estimate the individual effect of meteorological variables (vignette 2). a,** Simulated incidence rate ($C_t$) in Bogotá, Colombia (left panel) and Lübeck, Germany (right panel). Model parameters: basic reproduction number of 1.25, average duration of immunity of 1 year and other parameters as in Supplementary Table 3. In each panel, the black line shows the dynamic from the deterministic transmission model. The grey lines represent simulations (10 displayed out of 100 overall) from a stochastic transmission model, in which noise was added to the transmission rate at every time point (see Methods for full details). We implemented this stochastic variant only for this vignette to add realism in the form of model misspecification during estimation. **b,** Distribution of the 100 maximum likelihood estimates of the effect of temperature (left panel) and relative humidity in Bogotá and Lübeck. These estimates were obtained by direct maximization of the log-likelihood (that is, trajectory matching) by fitting the misspecified model—in which the transmission model was deterministic and the observation model stochastic—to data generated from the fully stochastic model—in which both the transmission and the observation models were stochastic. The dashed vertical lines indicate the true parameter value (−0.2) fixed in all model simulations. See Methods for further details about the estimation procedure.

leveraged to isolate the effects of temperature and relative humidity. To test this hypothesis, we used our illustrative model to run a range of numerical experiments in a pair of locations where this contrast was marked: Lübeck, Germany (53.9° N latitude, coefficient of variation (CV; standard deviation/mean) of temperature CV(Te) = 0.59, CV(RH) = 0.07, $r_s$(Te, RH) = −0.48) and Bogotá, Colombia (4.7° N latitude, CV(Te) = 0.04, CV(RH) = 0.07, $r_s$(Te, RH) = −0.10). By back-fitting our transmission model to 100 replicate time series of observed incidence rates it generated, we gauged how well we could estimate the effects of temperature and relative humidity (as well as other model parameters that would be unknown in real-world applications; Methods) in the two climates. In Lübeck, as expected for a climate characterized by low RH variability and large RH–Te correlation, the effect of temperature was estimated with more accuracy than that of relative humidity (mean AB of 0.02 and 0.09, respectively; Fig. 4). Of note, these results are reminiscent of those of vignette 1, except that the parameters estimated in this vignette originated from the true causal transmission model (Fig. 2) and, therefore, did not suffer from measurement bias. In Bogotá and its climate with low Te variability and almost null RH–Te correlation, the opposite result held, with higher accuracy for relative humidity than for temperature (mean AB of 0.04 and 0.05, respectively).

This simulation study thus suggests the scope for strategic choices regarding a study's location, where the local climate's properties can help estimate the effect of the weather variable of causal interest. Whenever more data are available, an alternative strategy is to use multilevel models, which provide a principled way to pool information while modelling variation across multiple locations[84]. Multilevel extensions are now routine for standard regression models but more challenging for the complex—typically nonlinear, stochastic and partially observed[92]—models needed to capture infectious disease dynamics. Nevertheless, recent statistical advances permit the estimation of such multilevel models[93,94], opening an avenue for large-scale dynamical modelling studies that harness information from multiple natural experiments in different climates. More broadly, this vignette underscores the critical importance of considering the causal mechanisms underlying weather dynamics.

**Vignette 3 on confounding bias and how climate variability can masquerade as spatial spread.** Spatial heterogeneities are commonly observed for infectious diseases[95–99]. Such heterogeneities can result from two broad classes of mechanisms, depending on whether they involve spatial interactions through the movement of individuals

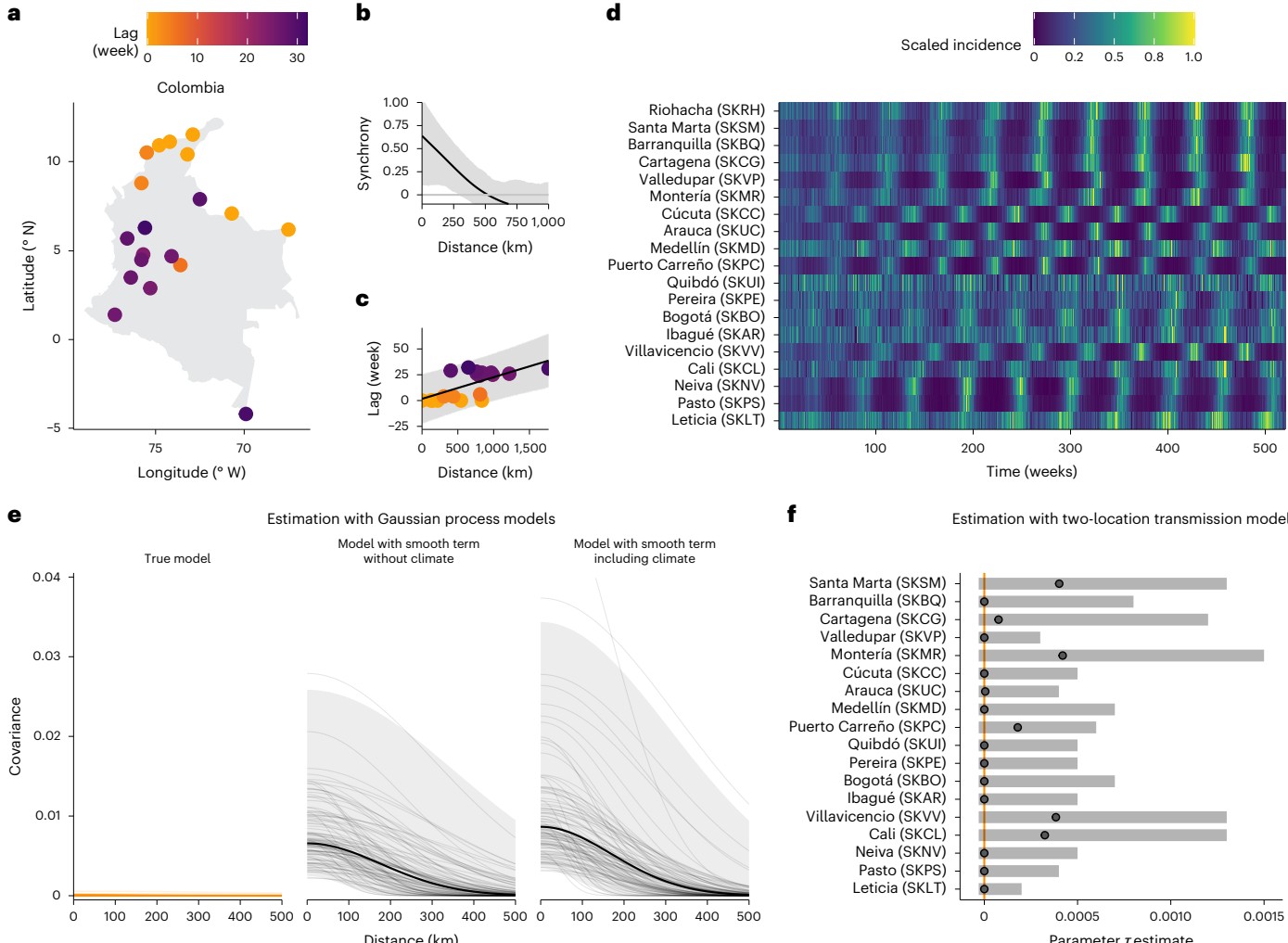

**Fig. 5 | Confounding bias and how climate variability can masquerade as spatial diffusion in Colombia (vignette 3). a,d**, Considering 19 locations across Colombia (**a**), we simulated incidence with no spatial diffusion but a common effect of climate on transmission (**d**; see Methods for full details). **b**, Pairwise epidemic synchrony between locations, with the fitted covariance function shown as a line and the 95% confidence interval shaded in grey. **c**, Relative timing of the simulated epidemic peaks across Colombia, referenced to the northernmost location (Riohacha). The colour indicates the time difference between the epidemic peaks of each site and those of the reference site. The line represents the line posterior mean from a Bayesian linear model, from which we estimated the travelling wave, and the grey envelope represents the 95% credible interval. **e**, Estimated spatial spread from Gaussian process models, with the dark lines representing the estimated mean covariance function, light lines representing 100 draws from the posterior distribution and shaded envelopes (orange and grey) indicating the 95% credible interval. **f**, Estimated spatial spread from two-location transmission models assuming no effect of climate, where the orange line denotes the true value of spatial spread, $\tau = 0$. The points represent the maximum likelihood estimate of the spatial spread, and the grey intervals represent the 95% confidence interval. Model parameters: basic reproduction number of 2.5, average duration of immunity of 2 years and other parameters as in Supplementary Table 3.

(that is, spatial spread) or spatial variation in some other variable (for example, climate)[100]. For example, spatial variations in climate across multiple locations may result in spatial covariance between these locations, even in the absence of spatial spread. This common cause of spatial variability may thus result in spurious associations that confound the estimated effect of spatial spread on observed incidence—that is, confounding bias (Fig. 1; see also Supplementary Fig. 3 for a DAG illustrating the problem in two locations).

To illustrate, we considered a scenario with seasonal transmission forced by weather but no spatial spread in two distinct countries (Supplementary Table 2): one with a definite latitudinal gradient in climate (Colombia)[101] and another with little spatial variability in climate (Spain). We simulated the resulting dynamics of our transmission model and assessed epidemic synchrony[102] across various locations in these two countries. In Colombia, the simulated incidence displayed diverse seasonal patterns that followed a latitudinal gradient broadly

matching that of the climate (Fig. 5b). In contrast, the low climatic variability in Spain resulted in tightly synchronous epidemics across the locations (Supplementary Fig. 4b). Hence, despite the absence of mechanisms causing spatial spread in our model, the shared effect of climate between locations resulted in marked spatial correlation in observed incidence, up to ~250 km in Colombia and more extensively throughout Spain (Fig. 5d and Supplementary Fig. 4d).

To further characterize this spatial correlation, we estimated the speed of the—spurious—travelling wave under the incorrect assumption of spatial spread being the sole cause of spatial heterogeneity[96]. Speed estimates were near infinite in Spain (Supplementary Fig. 4a,c), suggesting either confounding by climate or extremely strong coupling between the locations[74]. In Colombia, however, the speed was estimated at 218 km per month (95% credible interval (CI): 121–444 km per month; Fig. 5a,c), a value consistent with that documented for real travelling waves—for example, 110–320 km per month for pertussis

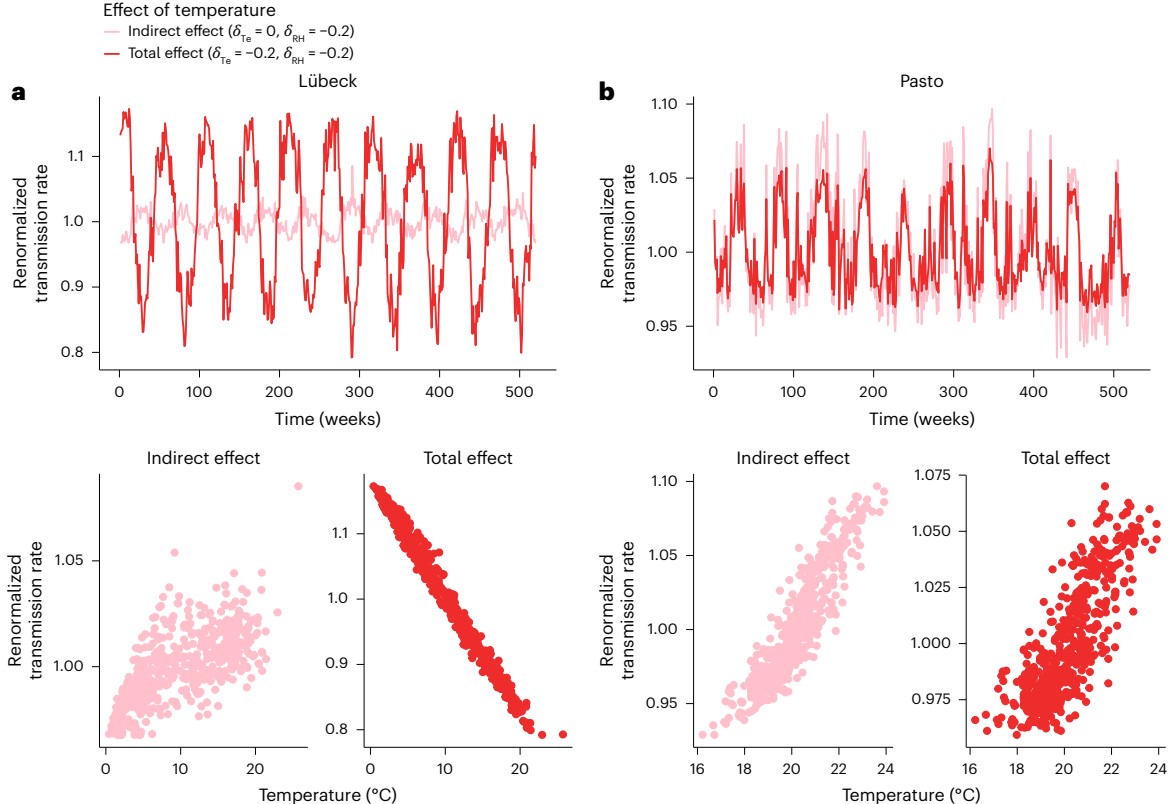

**Fig. 6 | Mediation and the direct and indirect causal effects of temperature on transmission (vignette 4). a,b,** Simulated transmission rate (top panels) in Lübeck, Germany (**a**) and Pasto, Colombia (**b**), from models including the total effect of temperature (dark lines, where the direct effect of temperature, $\delta_{Te}$, and the direct effect of RH, $\delta_{RH}$, were −0.2) or only the indirect effect of temperature mediated through humidity (light lines, where $\delta_{Te} = 0$ and $\delta_{RH} = -0.2$; see Methods for full details). The bottom panels show the indirect effect of temperature through relative humidity on transmission (left panels) and the total effect of temperature (right panels). Model parameters: basic reproduction number of 1.25, average duration of immunity of 1 year and other parameters as in Supplementary Table 3.

during 1951–2010 in the United States[98] and 150 km per month for dengue during 1983–1997 in Thailand[103].

To assess whether a statistical approach can address this confounding bias, we fitted a series of Gaussian process models to estimate the covariance in observed incidence rates between all locations in each country (Methods). As a control, we first tested the true model implied by the DAG, which included the sizes of the susceptible and infected populations, in addition to 1-week-lagged weather variables as covariates. As expected, this model correctly identified the absence of spatial spread, with the maximum covariance between locations being negligible ($\eta$ (95% CI) = 0.01 (0–0.02) in Colombia and 0 (0–0.02) in Spain; Fig. 5e and Supplementary Fig. 4e left panel). Next, we fitted models with smooths of time to try to capture the variations in the susceptible and infected population sizes, which would be unobserved in practice. We found that a model omitting weather variables (that is, the confounded model) estimated a spurious spatial covariance between the locations ($\eta$ (95% CI) = 0.09 (0.05–0.16) in Colombia and 0.38 (0.22–0.70) in Spain; Fig. 5e and Supplementary Fig. 4e middle panel). Furthermore, a model including the weather variables also led to biased estimates of spatial covariance ($\eta$ (95% CI) = 0.11 (0.06–0.18) in Colombia and 0.39 (0.23–0.74) in Spain; Fig. 5e and Supplementary Fig. 4e right panel). This finding thus re-emphasizes the difficulty of analysing incidence data because of measurement bias (vignette 1) and suggests the need for explicit models for capturing the unobserved variables.

Next, we thus designed a simple transmission model with climate forcing and spatial spread (described by a single coupling parameter $\tau$) between two locations. Using the data generated from the model with no spatial spread (Fig. 5d), we estimated the value of the coupling parameter (Methods). Here, the model including the weather variables—and thus controlling for the shared weather between the two locations—correctly revealed the absence of spatial spread (Fig. 5f and Supplementary Fig. 4f, with all confidence intervals of the maximum likelihood estimate for $\tau$ including 0). Although used for illustration here, this deliberately simple model could be extended to include more realistic features of spatial spread, such as multiple locations with different population sizes, potentially resulting in source–sink dynamics between urban and rural areas[95]. More generally, this vignette highlights the importance of integrating explicit causal models with transmission models to disentangle the mechanisms underlying spatial heterogeneities.

**Vignette 4 on mediation and the direct and indirect causal effects of temperature on transmission.** Weather is the consequence of a complex web of causally related environmental variables that can affect pathogen transmission through multiple pathways[65]. For instance, as illustrated in our simple causal graph and supported by experimental evidence[18], both temperature and relative humidity may directly affect transmission. However, because temperature also impacts relative humidity, its total causal effect may comprise a direct effect (Te → $\beta$) and an indirect effect mediated by relative humidity (Te → RH → $\beta$). Importantly, but counter-intuitively, the direct and indirect effects may act in opposite directions depending on the causal relationship between the parent variable (for example, temperature) and its mediator (for example, relative humidity; Fig. 1) in the environmental model.

To illustrate how these two effects play out in different settings, we simulated our model in Lübeck, Germany, and Pasto, another

Colombian city with relative humidity variability higher than in Bogotá. Owing to climatic differences causing temperature to be more variable than relative humidity in Lübeck (CV(Te) = 0.59, CV(RH) = 0.07) but less variable in Pasto (1.4° N latitude, CV(Te) = 0.07, CV(RH) = 0.19), we hypothesized the total impact of temperature would differ between the two cities. In each city, we considered two scenarios. In the first scenario, we set the two climatic parameters (see legend of Fig. 2) to their baseline values to capture the total effect of temperature—that is, the direct effect and the indirect effect mediated by relative humidity. In the second scenario, we set the climatic parameter of temperature to 0 to capture only its indirect effect, mediated by relative humidity.

In both cities, because of a combination of two negative effects (of temperature on relative humidity and relative humidity on transmission), higher temperature increased transmission through the indirect pathway ($r_s$(Te, $\beta$) = 0.91 in Pasto and 0.72 in Lübeck; Fig. 6a,b). In Lübeck, however, the higher temperature variability caused the direct effect to outweigh this indirect effect so that, overall, transmission decreased as temperature increased ($r_s$(Te, $\beta$) = −0.99; Fig. 6a). By contrast, the higher variability in relative humidity reversed the total effect in Pasto, where transmission increased with temperature ($r_s$(Te, $\beta$) = 0.80; Fig. 6b). Despite this overall positive effect, adjusting for relative humidity revealed the negative direct effect of temperature in Pasto (partial Spearman's rank correlation coefficient: $r_s$(Te, $\beta$|RH) = −0.56; Fig. 6b). Of note, echoing the results of vignette 1, the effects of temperature on the observed incidence rates were less definite because of measurement bias (Supplementary Fig. 5). Hence, despite identical causal mechanisms, climatic differences resulted in divergent effects of temperature in the two cities.

These conceptual insights have practical implications for interpreting the association between environment and transmission rates. Specifically, when evaluating the effect of temperature in a DAG similar to that in Fig. 2, models adjusting for relative humidity would identify the direct effect of both variables. In contrast, models without adjustment would only identify the total effect of temperature. The lack of clear causal frameworks may thus lead to misinterpreting model outputs, a risk described by earlier research as the 'table 2 fallacy'[104]. Hence, this vignette re-emphasizes the critical importance of causal reasoning and careful interpretation when probing the effect of climate on infection dynamics.

## Discussion

Here we aimed to show how causal inference concepts—such as descendants and mediators, confounding and measurement biases, and quasi-experiments—can guide research into the effects of climate on infectious diseases. Through a series of case studies, we illustrated how such concepts could help assess study design (vignette 1); strategically choose a study's location to achieve the set-up of a natural experiment (vignette 2); evaluate the risk of confounding bias (vignette 3); and interpret the direct and—sometimes paradoxical—indirect effects of meteorological variables on transmission (vignette 4). In addition, we showed that transmission models offer a principled and parsimonious tool to capture infectious disease dynamics and encapsulate causal frameworks. More broadly, seconding earlier calls in the epidemiological field[27], we argue that such frameworks are necessary for inferring the effect of weather and subsequently predicting the consequences of climate change on infectious diseases.

Because of this study's conceptual focus, we sidestepped the many methodological technicalities that inevitably arise in practice. In addition to mere data problems (for example, a mismatch between the time scales of observed data and infection dynamics), the effect of weather on infectious diseases can be more intricate than our simple model suggests. First, variables other than temperature and relative humidity may directly affect transmission, such that disentangling their direct and indirect effects may be more challenging than in vignette 4. For example, if one assumes a direct effect of dew point temperature on

transmission in the DAG represented in Fig. 2, then a causal mediation analysis would be required to estimate these different effects for all the environmental variables. Second, because of interindividual variability in the period separating exposure from infectiousness (that is, the latent period), the effect of weather on transmission is expected to be lag-distributed, resulting in a more complex causal diagram than Fig. 2. In this case, a standard compartmental model diagram may be more suited to depict the causal processes, with the added benefit that such diagrams can explicitly represent interactions and interference[82]. Third, this effect may be non-continuous and non-monotonic, as illustrated by recent experimental evidence showing a V-shape threshold association between relative humidity and survival time of coronaviruses[18,105]. Fourth, although the causal graph of Fig. 2 is realistic for pathogens causing acute, directly transmitted infections (such as respiratory viruses), the weather may have multiple effects on pathogens with more complex relationships with their human hosts. Such multiple effects are, for example, expected for invasive bacteria with prolonged carrier states (like the pneumococcus[106]), with weather affecting not only transmission of carriage but also progression from carriage to invasive disease[107,108]. Finally, the poor correlation between indoor and outdoor meteorological variables (especially observed for temperature and relative humidity[109]) has led to discussions about which measure of weather is more appropriate for causal inference, with indoor data argued to represent the bulk of weather exposures[16,110]. This problem may be viewed as another form of measurement bias and treated in a causal framework by modelling the causal link between indoor variables and their outdoor counterparts, with some recent research in this direction[110]. More generally, our simple causal framework could be similarly extended to tackle the other complexities listed above, as well as the potential biases illustrated in the vignettes simultaneously.

In conclusion, the expanding field of causal inference offers opportunities to strengthen evidence derived from observed data. This study thus presents an early effort to integrate this field with infectious disease epidemiology and climatology, with the ultimate research aims of elucidating how climate affects pathogens and predicting the consequences of climate change. Given that climate is an ever-present component of the environment, this research will also advance our understanding of the ecology of infectious diseases.

## Methods

### Model formulation

**Meteorological model.** The daily average records of dew point temperature (Td, expressed in °C) and ambient temperature (Te, expressed in °C) were extracted using the WeatherData function in Mathematica[111]. The data covered the period 2013–2022 in multiple weather stations located near the major cities of Colombia, Spain and Germany (Supplementary Table 1). Because the transmission model had a time step of 1 week, we calculated weekly averages of Te and Td for inclusion as covariates in the model. In case of missing daily records for a given week, we calculated the weekly average based on the records observed within that week in other years.

The relative humidity (RH, defined as the actual amount of water moisture in the air compared with the total amount that the air can hold at a given temperature) was then calculated using the formula[77,78]:

$$\log RH = \frac{\alpha Td}{\lambda + Td} - \frac{\alpha Te}{\lambda + Te}$$

where $\alpha$ = 17.625 and $\lambda$ = 243.04 °C are the revised Magnus coefficients[112]. This association can be understood intuitively as follows: relative humidity increases as the absolute moisture in the air (quantified by Td) increases, while it decreases as ambient temperature increases (because of the physical property that the maximum moisture air can hold increases exponentially with temperature). To verify the adequacy

of this formula, we also extracted actual RH records from the weather stations: the agreement between predictions and measurements was excellent (>90% correlation in all the locations considered), even though the predicted RH was overestimated at low ambient temperatures in temperate climates, such as in Germany.

In the following, we denote by $Te_t$ and $RH_t$ the weekly time series of ambient temperature and relative humidity, $\overline{Te}$ and $\overline{RH}$ their temporal averages (over the entire time series), and $Te'_t = \frac{Te_t}{\overline{Te}}$ and $RH'_t = \frac{RH_t}{\overline{RH}}$ their renormalized values.

**Transmission model.** To illustrate the different causal inference concepts, we formulated a discrete-time SIRS[74] model with the transmission rate $\beta_t$ forced by the two climatic variables:

$$\log \beta_t = \log \beta + \delta_{Te}(Te'_t - 1) + \delta_{RH}(RH'_t - 1)$$

where $\beta$ is the average transmission rate, $\delta_{Te}$ the effect of ambient temperature and $\delta_{RH}$ the effect of relative humidity. Based on experimental evidence on respiratory viruses[18,113], we assumed a small negative effect of both climatic variables: $\delta_{Te} = \delta_{RH} = -0.2$. In other words, we assumed that transmission decreased as either climatic variable increased.

To derive the equations of the discrete-time model, we first write the system of ordinary differential equations for the continuous-time model:

$$\frac{dS}{dt} = \mu N + \alpha R - (\lambda(t) + \mu)S$$

$$\frac{dI}{dt} = \lambda(t)S - (\gamma + \mu)I$$

$$\frac{dR}{dt} = \gamma I - (\alpha + \mu)R$$

where $\lambda(t) = \beta(t)I(t)/N$ is the force of infection, $N$ is the population size, $\mu$ is the birth/death rate, $\gamma^{-1}$ is the generation time, and $\alpha^{-1}$ is the average duration of protection. All parameters are listed in Supplementary Table 3. Assuming a fixed time step $\Delta t = 1$ week and a fixed generation time equal to this time step ($\gamma^{-1} = \Delta t = 1$ week), we discretized the system of ordinary differential equations (using the approximation $\frac{dX}{dt} \approx \frac{X_{t+\Delta t} - X_t}{\Delta t}$) to get the equations of the discrete-time model:

$$S_{t+1} = S_t + \mu N + \alpha R_t - (\lambda_t + \mu)S_t$$

$$I_{t+1} = \lambda_t S_t - \mu I_t$$

$$R_{t+1} = R_t + I_t - (\alpha + \mu)R_t$$

For all simulations, we initialized the state variables to their equilibrium values for the model with no seasonal forcing ($\delta_{Te} = \delta_{RH} = 0$). For the discrete-time model, these equilibria are given by $S^* = \frac{N}{R_0}$, $I^* = \frac{\alpha + \mu}{\alpha + \mu + 1}(N - S^*)$ and $R^* = N - S^* - I^*$, where $R_0 = \frac{\beta}{\mu + 1}$ is the basic reproduction number.

**Observation model.** To complete the model formulation, we specified a stochastic observation model to generate observed data from the transmission model's outputs. Let

$$C_t = \lambda_{t-1} S_{t-1} = \beta_{t-1} \frac{I_{t-1}}{N} S_{t-1}$$

represent the (true) incidence rate (Fig. 2), defined as the weekly number of new cases. We then used a negative binomial (NB) model to sample the observed incidence rate:

$$C_t^{(O)} \sim NB(\mu_t = \bar{\rho} C_t, \rho_k)$$

where $\bar{\rho}$ is the mean reporting probability and $\rho_k$ the reporting over-dispersion, representing extra variability in the mean reporting probability.

**Complete model and causal graph.** The complete model thus consisted of the discrete-time transmission model and the observation model described above. Because of our simplifying assumptions (in particular, a fixed generation time and an immediate effect of climate on transmission), this model was exactly represented by the causal graph displayed in the main text (Fig. 2).

**Numerical implementation.** The model was implemented in the R package pomp[114], operating in R version 4.4.1[115]. All figures were created with the R package ggplot[116], and the data for the maps was obtained from Natural Earth (https://www.naturalearthdata.com). Other packages used for specific vignettes are cited below.

**Vignette 1 on descendants and measurement bias**
**Simulation details.** The simulations for this vignette were based on climatic data in Lübeck, Germany, a location with a temperate oceanic climate (Köppen–Geiger classification: Cfb) characterized by large seasonal variability in temperature (CV: 0.59), little variability in relative humidity (CV: 0.07) and marked correlation between the two ($r_s = -0.48$). The model was simulated for three different values of the basic reproduction number (1.25, 2.5 and 5) and an average duration of immunity of 1 year; the other parameters were fixed to the values indicated in Supplementary Table 3.

**Regression model for time series of observed cases.** To identify a candidate regression model for the variable $C_t^{(O)}$ (observed incidence rate), we first log-transformed the variable $C_t$ (true incidence rate):

$$\log C_t = \log\left(\beta_{t-1} \frac{I_{t-1}}{N} S_{t-1}\right) = C + \delta_{Te} Te'_{t-1} + \delta_{RH} RH'_{t-1} + \log(I_{t-1} S_{t-1})$$

where $C$ is a constant. This equation, alongside the negative binomial observation model connecting $C_t^{(O)}$ and $C_t$, shows that a natural candidate model for $C_t^{(O)}$ is a negative binomial regression model with log-link, and $Te'_{t-1}$ and $RH'_{t-1}$ as covariates. In practice, the variable $I_{t-1}S_{t-1}$ is unobserved but may be captured by including a function of time as a covariate[68]. Here, we fitted a negative binomial GAM[117] that included the climatic covariates and a smooth of time to capture temporal variations in this variable. To give the regression model enough flexibility to capture these variations (which may occur over fast time scales; see Fig. 3), we set the basis dimension of the smooth to 50, or approximately 1 degree of freedom per 10 weeks of data. As a control, we also verified that the exact model with $\log(S_{t-1}I_{t-1})$ as a covariate yielded, on average, unbiased estimates of $\delta_{Te}$ and $\delta_{RH}$ (Supplementary Fig. 1). All the regression models were fitted using the mgcv package (version 1.8-42) in R[117].

**Regression models for time series of effective reproduction numbers.** By definition, the time-varying effective reproduction number $R_{e,t}$ equals:

$$R_{e,t} = \frac{R_{0,t} S_t}{N} \approx \frac{\beta_t S_t}{N}$$

where $R_{0,t} = \frac{\beta_t}{(\mu + 1)}$ is the time-varying basic reproduction number and $\mu \ll 1$ (per week) is the death rate. Taking logs, we find that:

$$\log R_{e,t} = C + \delta_{Te} Te'_t + \delta_{RH} RH'_t + \log(S_t)$$

where $C$ is a constant. The corresponding DAG, shown in Supplementary Fig. 2, differs from the DAG for the incidence rate (Fig. 2) in that

the effective reproduction number depends on just one unobserved variable ($S_t$), while the incidence rate depends on two ($S_t$ and $I_t$).

To derive an estimator for $R_{e,t}$, we first express it as a function of the incidence rates $C_t$. Specifically (see 'Model formulation'), $C_{t+1} = \lambda_t S_t = \frac{\beta_t I_t S_t}{N} \approx R_{e,t} I_t$ and $I_t = C_t - \mu I_{t-1} \approx C_t$ (neglecting mortality). Hence, we find the following equation for $R_{e,t}$:

$$R_{e,t} \approx \frac{C_{t+1}}{C_t}$$

In other words, the effective reproduction number is approximately equal to the epidemic growth rate, as expected intuitively because of our assumption of a 1-week fixed generation time.

In practice, because of under-reporting, the true values of $R_{e,t}$ are unobserved. However, estimation is possible via renewal models that back-calculate $R_{e,t}$ values from the generation time distribution and the observed incidence rates. Denoting by $R_{e,t}^{(O)}$ this estimator, we generated it as follows:

$$\log R_{e,t}^{(O)} \sim N(\mu = \log R_{e,t}, \sigma = 0.1)$$

In other words, we assumed a small dose of noise (approximately 10% around the true value) but no systematic bias in the estimation $R_{e,t}$.

We then fitted normal ($N$) regression models of $\log R_{e,t}^{(O)}$ (outcomes) with $\text{Te}'_t$, $\text{RH}'_t$ and a smooth of time as covariates (Supplementary Fig. 1). As a control, we also verified that the true regression model with $\text{Te}'_t$, $\text{RH}'_t$ and $\log S_t$ as covariates yielded, on average, unbiased estimates (Supplementary Fig. 1).

### Vignette 2 on climate variability as natural experiments

**Simulation details.** The simulations for this vignette were based on climatic data in Bogotá, Colombia, and Lübeck, Germany. In marked contrast to Lübeck's climate (see above), Bogotá's climate is classified as warm and temperate (Köppen classification: Csb), with little seasonality in temperature because of proximity to the Equator (CV: 0.04) but larger variability in relative humidity (CV: 0.07) and decoupling between the two climatic variables ($r_s = -0.1$).

To introduce model misspecification during the estimation of model parameters, we implemented a stochastic transmission model where the deterministic transmission rate $\beta_t$ was multiplied at every time step by gamma white noise with mean 1 and standard deviation 0.02 (ref. [118]). The complete model was, therefore, fully stochastic, with noise affecting both the transmission and the observation models. The model parameters were set as follows: basic reproduction number of 1.25, average duration of immunity of 1 year and other parameters fixed to the values indicated in Supplementary Table 3.

**Parameter estimation protocol.** The following six model parameters were assumed unknown and estimated from the data: basic reproduction number ($R_0$), waning immunity rate ($\alpha$), mean reporting probability ($\bar{\rho}$), reporting over-dispersion ($\rho_k$) and climatic parameters ($\delta_{Te}$ and $\delta_{RH}$). To generate the synthetic data for estimation, we first generated 100 replicate time series of observed weekly cases ($C_t^{(O)}$) from the fully stochastic model. For every replicate time series, we then fitted the misspecified model—with a deterministic transmission model and stochastic observation model—using trajectory matching[92]. Specifically, we used the Nelder–Mead algorithm[119] (initialized at the true parameter values) to maximize the log-likelihood and identify the maximum likelihood parameter estimates. All the parameters were estimated on an unconstrained scale using log (parameters $R_0$, $\alpha$ and $\rho_k$) or logit (parameter $\bar{\rho}$) transformations.

### Vignette 3 on confounding bias

**Simulation details.** The simulations for this vignette were based on climatic data from 15 weather stations in Spain and 19 in Colombia,

located near the major cities of both countries. Continental Spain exhibits relatively uniform temperate climates with consistent seasonal variations of temperature and humidity seasonality across the country. In contrast, Colombia displays a range of tropical climates, with diverse seasonal patterns of precipitation along a latitudinal gradient[101]. The models were simulated without incorporating spatial spread between the locations and with a basic reproduction number of 2.5 and an average duration of immunity of 2 years to achieve yearly epidemics in all the locations. The other parameters were fixed to the values indicated in Supplementary Table 3.

**Epidemic synchrony and assessment of spatial spread.** To assess spatial synchrony between locations, we estimated the non-parametric (cross-) correlation function (NCF) of the simulated time series using the NCF package[100]. We estimated the 95% confidence intervals using 500 bootstraps.

To diagnose the risk of confounding bias caused by climate, we estimated the speed of the potential travelling wave that would have resulted from spatial diffusion. To do so, we first estimated the difference in epidemic peak timing as the lag (in weeks), maximizing the cross-correlation function between every location and a reference location (Riohacha for Colombia and Gijón for Spain). We then regressed this difference against the geographical distance between locations and estimated the speed wave as the inverse of the regression coefficient.

To further evaluate how climate can confound the estimate of spatial spread between locations, we fitted negative binomial Gaussian process models, that assumes a joint multivariate normal (MVN) distribution, to the time series of observed incidence from all locations in Colombia:

$$\begin{pmatrix} \log C_t^{(O)(1)} \\ \log C_t^{(O)(2)} \\ \dots \\ \log C_t^{(O)(19)} \end{pmatrix} \sim \text{MVN} \left( \begin{bmatrix} \log \mu_t^{(1)} \\ \log \mu_t^{(2)} \\ \dots \\ \log \mu_t^{(19)} \end{bmatrix}, K \right)$$

The Gaussian process captures the covariance $K^{(ij)}$ between the incidence of any pair of locations $i$ and $j$ as a function of the geographic distance between locations, defined by the exponential quadratic kernel:

$$K^{(ij)} = \eta^2 \exp\left( -\frac{||x^{(i)} - x^{(j)}||^2}{2\varphi^2} \right)$$

where $||x^{(i)} - x^{(j)}||$ is the Euclidean distance between two locations, $\eta$ is the maximum covariance between locations and $\varphi$ is a characteristic distance that controls the spatial scale over which the covariance varies. Based on the proposed 'true' and 'smooth' regression models described in the methods for vignette 1, we fitted models including the size of the infected and susceptible population or a smooth of time with or without the environmental variables. For example, for the true model, we wrote $\log \mu_t^{(1)} \approx \log(S_{t-1}^{(1)} I_{t-1}^{(1)}) + \text{Te}_{t-1}^{(1)'} + \text{RH}_{t-1}^{(1)'}$ in location 1. Finally, we calculated the correlation matrix between the incidence of all locations from the estimated covariance matrix $K$. All models were fitted using the brms package[120] with four Markov chains, each run for 10,000 iterations, assuming uninformative priors.

**Two-location transmission model with spatial diffusion.** As purely statistical approaches resulted in confounded estimations of spatial spread, we moved on to estimate spatial spread with transmission models. We extended our climate-forced SIRS model to include spatial diffusion by dividing the population into two coupled locations, for $i$ = 1,2 (refs. [121],[122]):

$$S_{t+1}^{(i)} = S_t^{(i)} + \mu N^{(i)} + \alpha R_t^{(i)} - (\lambda_t^{(i)} + \mu) S_t^{(i)}$$

$$I_{t+1}^{(i)} = \lambda_t^{(i)} S_t^{(i)} - \mu I_t^{(i)}$$

$$R_{t+1}^{(i)} = R_t^{(i)} + I_t^{(i)} - (\alpha + \mu) R_t^{(i)}$$

with the force of infection in each location $i$ given by:

$$\lambda^{(i)}(t) = \frac{\beta^{(i1)} I^{(1)}(t)}{N^{(1)}} + \frac{\beta^{(i2)} I^{(2)}(t)}{N^{(2)}}$$

We assumed a symmetric scenario where the transmission rate is the same within patches ($\beta^{(ij)} = \beta$ if $i = j$) but differs *between* patches ($\beta^{(ij)} = \tau\beta$ if $i \neq j$, where $0 \leq \tau \leq 1$ is the coupling strength). In this case, each location had the same population ($N^{(1)} = N^{(2)} = N$), and the endemic equilibria were independent of the location, given by $S^* = \frac{N}{R_0}$, $I^* = \frac{\alpha + \mu}{\alpha + \mu + 1}(N - S^*)$, and $R^* = N - S^* - I^*$, where $R_0 = \frac{\beta(1+\tau)}{\mu+1}$.

**Parameter estimation protocol.** As in vignette 2, we assumed the same six parameters unknown and estimated from the simulated data without spatial spread. We fitted the two-location transmission model with spatial diffusion for every location and a reference location (Riohacha for Colombia and Gijón for Spain) using trajectory matching[92]. Again, we used the Nelder–Mead algorithm[119] (initialized at the true parameter values) to maximize the log-likelihood and identify the maximum likelihood parameter estimates. Then, we generated likelihood profiles to estimate the spatial spread parameter $\tau$ by varying the parameter while maximizing the likelihood over the remaining parameters to obtain likelihood-ratio-test-based confidence intervals. All the parameters were estimated on an unconstrained scale using log (parameters $R_0$, $\alpha$ and $\rho_k$) or logit (parameter $\bar{\rho}$ and $\tau$) transformations.

### Vignette 4 on mediation, direct and indirect causal effects
**Simulation details.** The simulations in this vignette were based on climatic data from Pasto, Colombia, and Lübeck, Germany. As described before, in Lübeck, temperature displays larger seasonal variability than relative humidity. In contrast, in Pasto, seasonal variability in relative humidity is larger (CV: 0.19) than in temperature (CV: 0.07).

As temperature affects humidity, the total causal effect of temperature comprises a direct effect and an indirect effect mediated by relative humidity. Thus, we simulated a model representing the total effect of temperature (with climatic parameters to $\delta_{Te} = -0.2$ and $\delta_{RH} = -0.2$) and another model representing only the indirect effect of temperature (with climatic parameters fixed to $\delta_{Te} = 0$ and $\delta_{RH} = -0.2$). For these simulations, we fixed the other parameters as follows: basic reproduction number of 1.25, average duration of immunity of 1 year and other parameters as indicated in Supplementary Table 3.

### Reporting summary
Further information on research design is available in the Nature Portfolio Reporting Summary linked to this article.

## Data availability
All data are available at https://github.com/DomenechLab/Causality_Seasonality and stored in Edmond, the open research data repository of the Max Planck Society, at https://doi.org/10.17617/3.9CWN7W. All weather data were extracted using the WeatherData function in Mathematica (https://reference.wolfram.com/language/ref/WeatherData.html), with the weather station names indicated in Supplementary Table 2.

## Code availability
All R programming codes are available at https://github.com/DomenechLab/Causality_Seasonality and stored in Edmond, the open research data repository of the Max Planck Society, at https://doi.org/10.17617/3.9CWN7W.

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

## Acknowledgements

This work was funded by the Max Planck Society through the core funding of M.D.d.C.'s Max Planck Research Group at the Max Planck Institute for Infection Biology.

## Author contributions

L.A.B.G. and M.D.d.C. conceptualized the project and designed the methods. Model implementation and analyses were carried out by L.A.B.G. and M.D.d.C. L.A.B.G. and M.D.d.C. wrote the paper, with support from S.C.K. and T.K. M.d.d.C. supervised the project. All authors approved the final version of the paper.

## Funding

## Competing interests

T.K. reports outside the submitted work, having received research grants from the Gemeinsamer Bundesausschuss (G-BA—Federal Joint Committee, Germany). He has also received personal compensation from Eli Lilly and Company, Novartis, the BMJ and Frontiers. M.d.d.C. received postdoctoral funding (2017–2019) from Pfizer and consulting fees from GSK. The other authors declare no competing interests.

## Additional information

**Correspondence and requests for materials** should be addressed to Matthieu Domenech de Cellès.

# Reporting Summary

## Statistics

For all statistical analyses, confirm that the following items are present in the figure legend, table legend, main text, or Methods section.

| n/a | Confirmed | |
|---|---|---|
| ☐ | ☒ | The exact sample size (*n*) for each experimental group/condition, given as a discrete number and unit of measurement |
| ☐ | ☒ | A statement on whether measurements were taken from distinct samples or whether the same sample was measured repeatedly |
| ☐ | ☒ | The statistical test(s) used AND whether they are one- or two-sided<br>*Only common tests should be described solely by name; describe more complex techniques in the Methods section.* |
| ☐ | ☒ | A description of all covariates tested |
| ☐ | ☒ | A description of any assumptions or corrections, such as tests of normality and adjustment for multiple comparisons |
| ☐ | ☒ | A full description of the statistical parameters including central tendency (e.g. means) or other basic estimates (e.g. regression coefficient) AND variation (e.g. standard deviation) or associated estimates of uncertainty (e.g. confidence intervals) |
| ☒ | ☐ | For null hypothesis testing, the test statistic (e.g. *F*, *t*, *r*) with confidence intervals, effect sizes, degrees of freedom and *P* value noted<br>*Give P values as exact values whenever suitable.* |
| ☐ | ☒ | For Bayesian analysis, information on the choice of priors and Markov chain Monte Carlo settings |
| ☒ | ☐ | For hierarchical and complex designs, identification of the appropriate level for tests and full reporting of outcomes |
| ☐ | ☒ | Estimates of effect sizes (e.g. Cohen's *d*, Pearson's *r*), indicating how they were calculated |

*Our web collection on statistics for biologists contains articles on many of the points above.*

## Software and code

Policy information about availability of computer code

| | |
|---|---|
| Data collection | The simulated data was generated from a mathematical model of transmission using the package pomp (version 5.11) in R (version 4.4.1). The daily average records of dew point temperature (Td, expressed in °C) and ambient temperature (Te, expressed in °C) were extracted using the WeatherData function in Mathematica (version 12.0). |
| Data analysis | All analyses were conducted in R (version 4.4.1). For reproducibility, the renv package (version 1.0.7) was used to keep track of all the packages' version. The model was coded and run using the package pomp (version 5.11). Further analyses were performed using the packages mgcv (version 1.9-1), brms (version 2.21.0), and ncf (version 1.3-2). All code can be found at: https://github.com/DomenechLab/Causality_Seasonality and are stored in Edmond, the open research data repository of the Max Planck Society, https://doi.org/10.17617/3.9CWN7W. |

For manuscripts utilizing custom algorithms or software that are central to the research but not yet described in published literature, software must be made available to editors and reviewers. We strongly encourage code deposition in a community repository (e.g. GitHub). See the Nature Portfolio guidelines for submitting code & software for further information.

# Data

Policy information about availability of data

All manuscripts must include a data availability statement. This statement should provide the following information, where applicable:

- Accession codes, unique identifiers, or web links for publicly available datasets
- A description of any restrictions on data availability
- For clinical datasets or third party data, please ensure that the statement adheres to our policy

All data and R programming codes are available at https://github.com/DomenechLab/Causality_Seasonality and stored in Edmond, the open research data repository of the Max Planck Society, https://doi.org/10.17617/3.9CWN7W.

# Research involving human participants, their data, or biological material

Policy information about studies with human participants or human data. See also policy information about sex, gender (identity/presentation), and sexual orientation and race, ethnicity and racism.

| | |
|---|---|
| Reporting on sex and gender | NA |
| Reporting on race, ethnicity, or other socially relevant groupings | NA |
| Population characteristics | NA |
| Recruitment | NA |
| Ethics oversight | NA |

Note that full information on the approval of the study protocol must also be provided in the manuscript.

# Field-specific reporting

Please select the one below that is the best fit for your research. If you are not sure, read the appropriate sections before making your selection.

☐ Life sciences   ☐ Behavioural & social sciences   ☒ Ecological, evolutionary & environmental sciences

For a reference copy of the document with all sections, see nature.com/documents/nr-reporting-summary-flat.pdf

# Ecological, evolutionary & environmental sciences study design

All studies must disclose on these points even when the disclosure is negative.

| | |
|---|---|
| Study description | This study explores how concepts from causal inference can be applied to understand the effects of weather on the transmission and infection dynamics of infectious diseases in the context of climate change. We use mathematical models of transmission to generate simulations and demonstrate how these concepts improve study design, reduce bias, and enhance the interpretation of the impacts of climatic variables on disease transmission. |
| Research sample | We generated simulations from a mathematical model of transmission forced by weather (temperature and relative humidity). We used weather data from major cities of Colombia, Spain, and Germany. We chose these locations according to the characteristic of the climate. |
| Sampling strategy | This is a simulation study that used previously-collected climatic data, we did not conduct any sampling. |
| Data collection | The climatic data was collected from multiple weather stations located near the major cities of Colombia, Spain, and Germany. |
| Timing and spatial scale | The simulations were generated using weekly climatic data covering the period 2013–2022 in multiple weather stations located near the major cities of Colombia, Spain, and Germany. |
| Data exclusions | We did not excluded data for this study. |
| Reproducibility | All code to simulate the data and repeat the analyses has been made publicly available and the renv package (version 1.0.7) was used to keep track of all the packages' version. |
| Randomization | We simulated data from a mathematical model of transmission; no randomization was conducted. |
| Blinding | This is a modeling study, in which we used a transmission model to simulate data; blinding is not relevant to this study design. |

Did the study involve field work?   ☐ Yes   ☒ No

# Reporting for specific materials, systems and methods

We require information from authors about some types of materials, experimental systems and methods used in many studies. Here, indicate whether each material, system or method listed is relevant to your study. If you are not sure if a list item applies to your research, read the appropriate section before selecting a response.

## Materials & experimental systems

| n/a | Involved in the study |
|-----|------------------------|
| ☒ ☐ | Antibodies |
| ☒ ☐ | Eukaryotic cell lines |
| ☒ ☐ | Palaeontology and archaeology |
| ☒ ☐ | Animals and other organisms |
| ☒ ☐ | Clinical data |
| ☒ ☐ | Dual use research of concern |
| ☒ ☐ | Plants |

## Methods

| n/a | Involved in the study |
|-----|------------------------|
| ☒ ☐ | ChIP-seq |
| ☒ ☐ | Flow cytometry |
| ☒ ☐ | MRI-based neuroimaging |

## Plants

| | |
|---|---|
| Seed stocks | NA |
| Novel plant genotypes | NA |
| Authentication | NA |

