## [Peer Review File · Nature Ecology & Evolution]

Causal inference concepts can guide research into the effects of climate on infectious diseases

Corresponding Author: Dr Matthieu Domenech de Cellès

Version 0:

Decision Letter:

16th April 2024

Dear Dr Domenech de Cellès,

Your Perspective entitled "How causal inference concepts can guide research into the effects of climate on infectious diseases" has now been seen by three reviewers. You will see from their attached comments that they find your Perspective article of interest, but they have raised several points that need to be addressed before we can make a decision on publication. We will need to consider your response to these concerns in the form of a revised manuscript, accompanied by a separate list to explain your revisions.

We note that Reviewer 2 (and their co-reviewer) have recommended to rework the manuscript substantially or remove two of your vignettes. We don't feel that that is necessary, but would encourage you to still consider these reviewers' comments carefully, in terms of the 'thread' and narrative of the piece, and also trying to make it more accessible to readers not already well versed in this area. We do try to publish Perspective articles that are interesting and relevant to a broad readership.

We very much hope that you will be able to address these comments of our reviewers and would like to invite you to revise your manuscript accordingly.

A separate point-by-point response to ALL of the concerns raised by the reviewers should be included with the revised manuscript, in each case describing what changes have been made to the manuscript or, alternatively, if no action has been taken, providing a compelling argument for why that is the case.

When you are ready, please use the link below to submit the revised version (with the changes clearly marked).

Link Redacted

- include a point-by-point response to any editorial suggestions and to our reviewers. Please include your response to the editorial suggestions in your cover letter, and please upload your response to the referees as a separate document.

- ensure it complies with our format requirements for Perspectives as set out in our guide to authors at www.nature.com/natecolevolate/authors/gta/submit/index.html (More detailed guidelines can be found here: <https://mts-natecolevol.nature.com/letters/ReviewGuidelines.pdf>)

- state in a cover note the length of the text, methods and legends; the number of references and the number of display items (figures, tables and uncaptioned molecular structures).

Please ensure that all correspondence is marked with your Nature Ecology & Evolution reference number in the subject line.

Nature Ecology & Evolution is committed to improving transparency in authorship. As part of our efforts in this direction, we are now requesting that all authors identified as 'corresponding author' on published papers create and link their Open Researcher and Contributor Identifier (ORCID) with their account on the Manuscript Tracking System (MTS), prior to acceptance. ORCID helps the scientific community achieve unambiguous attribution of all scholarly contributions. You can create and link your ORCID from the home page of the MTS by clicking on 'Modify my Springer Nature account'. For more information please visit please visit <http://www.springernature.com/orcid>.

[redacted]

Reviewer Comments:

Reviewer #1 (Remarks to the Author):

This is a terrific paper, and very important. The authors' approach of simulating and then analyzing epidemics to identify methodological pitfalls is elegant. I would point out that while they refer to the studies of interest as observational, most of these studies use ecological exposure data which creates pitfalls, but also confers some strengths, and indeed opens up a longstanding environmental epidemiology toolbox as a useful array of approaches that can be applied to these problems.

I have a few relatively minor comments which I hope are useful to the authors.

1. I think DAGs are incredibly helpful in making models and assumptions explicit. To the issue of ecological exposures (ubiquitous exposures such as weather, air pollutants, UV radiation and so forth), the problem of ecological fallacy is well recognized. I think the fact that use of ecological exposures makes observational studies invulnerable to confounding by case characteristics (which a DAG makes clear...health status, age and sex can't change the weather) is less commonly recognized.

2. This is an interesting moment to be writing about climate change and models, because of course (in the pages of Nature no less) it has recently been noted that climate models themselves are beginning to fail, presumably as a result of as-yet unrecognized positive feedback loops in climatic systems. This may warrant mention. See: <https://www.nature.com/articles/d41586-024-00816-z>.

3. Seasonality is of course a ubiquitous attribute of communicable diseases (and it would be nice to cite what I think is now a bit of a landmark work on this by Micaela Martinez: <https://journals.plos.org/plospathogens/article?id=10.1371/journal.ppat.1007327>). But an important fix to this, which doesn't seem to be mentioned in the MS, is simply using smoothers (whether fast Fourier terms or cubic splines, or other trend terms) to eliminate nonspecific seasonal oscillation that can be driven by behaviour, immune effects or environmental effects. That means that we are effectively regressing on the RESIDUALS of environmental exposures...and we get coefficients interpretable as effects that attend deviation from expected seasonal oscillation. If we don't do this we get predictable "pirates prevented climate change" sorts of correlations...but I think the importance of separating smooth and rough components of time series using smoothers is pretty well recognized.

4. For climate change effects we are somewhat held back by the lack of a "control planet"...we only have one planet. This led scholars like Hsiang to use large scale, irregular oceanic phenomena like ENSO as "climate change-like shocks" to evaluate likely climate change effects. It seems to me that on a DAG ENSO might look and awful lot like a randomized exposure or instrumental variable...

5. I think there are other fixes to some of the issues the authors flag...the issue of multi-causality in infectious disease research and environmental exposures is pretty well recognized (e.g., with Lyme disease we can see precipitation effects at very long lags (mouse populations), and much shorter lags (human behaviour)). Here the use of the environmental epidemiology toolbox (e.g., distributed nonlinear lag models) can be really useful. Similarly, I think the issues around "pseudo-diffusion" that the authors flag would be readily addressed using multilevel models that treat geographies as fixed or random effects.

Reviewer #2 (Remarks to the Author):

Summary: The manuscript aims to raise awareness, and briefly introduce, some key ideas from the literature on causal inference. The target audience seems to be a mix of epidemiologists and climate scientists who are interested in the nexus between climate change and infectious disease. The authors assert that this audience is unaware of key ideas from the last twenty to thirty years from scholarship aimed at improving the quality of causal inferences from data (i.e., innovations aimed at strengthening causal claims from observational data). To illustrate a few of these ideas and highlight how understanding them can improve empirical inferences that scholar draw about the causal relationships between weather and disease, the authors use four simulated examples ("vignettes").

Reviewer Background: This review reflects a co-review by a social scientist and a natural scientist, both of whom study causal inference and have written articles that seek to communicate about causal inference methods to scholarly and practitioner audiences who are unfamiliar with these methods. One of us also has research experience in infectious disease, but neither of us works at the nexus of climate and disease. Thus, our review focuses on the accuracy of the text and our impressions of how likely the target audience for this manuscript will understand the concepts being presented, based on our experience communicating with scientists unfamiliar with causal inference frameworks. Given our background, we are not able to determine whether the ideas described in this manuscript are already well-known to researchers working at the climate-disease nexus. Our review assumes that the authors are correct that most researchers working in this area are not familiar with these ideas despite the fact that there are articles about similar ideas targeted at climate scientists (e.g., Runge et al., 2019, Nat Comm) and epidemiologists (many articles, including some that include a SIR/SIRS model; eg., Ackley et al., 2017, Epi Methods).

Main Comments

1. Errors: Apart from Vignette 1, we did not find any serious errors in the text.

2. Readability: Our main concern with the manuscript is whether the targeted audience will be able to understand the text. The

intent of the manuscript is to illustrate the value of viewing research questions in this field through the lens of a “causal inference framework.” Yet, the text doesn’t clarify what a “causal inference framework” is and, in all four vignettes, does not explain how readers are supposed to use such a framework to improve their analysis. If we (the reviewers) hadn’t been so familiar with the ideas being presented, we doubt that we would have been able to follow the text. We expect that, to the novice reader, the vignettes will read like a set of objects that the authors have pulled from a larger bag of objects that is never described. The ideas in the vignettes are indeed conceptually connected under popular causal inference frameworks, but the manuscript doesn’t make these connections clear in the text. So, we worry that the thread of the argument may be difficult for a reader from the target audience to follow. Ultimately, the best review on readability will be from members of the target audience who can say, “Yes, this information is new to me or would be to my collaborators, and, yes, I can understand what the authors are saying.” But, if we’re correct about the readability challenge of this manuscript, the value of the manuscript drops considerably. We can see two potential solutions that may help a bit.

- a. Completely revise to focus more on the stated aim: the value of causal inference frameworks (the constraints of the NE&E Perspective format may not be conducive to introducing these important ideas to a novice audience).
- b. Reduce some of the text and focus on one or two of the subjects of the vignettes rather than four of them (like 1 and 2, which are connected). And then do a better job of explaining the topics and highlighting better in the intro that these topics are neglected in the literature and that publications targeted at epidemiologists and climate scientists on the utility of causal inference frameworks have not covered these topics in a way that is relevant for climate-disease scientists. To find more space for explanations, move some text to the Supplement: the text from lines 99-140, which explains details of the models and the justifications for its features, and the text from lines 333-356, which highlights limitations of the model and the vignettes (in main text, you can just say that real estimation is more challenging and that’s all the more reason to use a clear causal framework, and then provide more details in Supp). Use the saved text to do a better job of explaining the key ideas and their relevance to the target audience.

Other Comments

1. Vignette 1:

- a. If the underlying (data-generation) procedure were exactly as shown in Figure 1 and described in Vignette 1, removing a term that includes S_t and/or I_t from the model should not result in biased effect estimators, since neither are shown or discussed as confounders in the figure or text. However, the transmission rate β_t is a function of the average transmission rate β (as given in the Supplementary Text), which itself is a function of R_0 and μ (according to the R code linked in the manuscript). The parameter μ influences the replenishment of the total population, thus influencing both S_t and I_t , which is likely where the confounding is being introduced. If this is indeed the case, none of these key details are reflected in Vignette 1 or Figure 1. Since transmission rates are usually estimated from prevalence and incidence data, it would be expected that S_t and I_t have some influence on β that is not reflected by Vignette 1. Given the manuscript’s intent to focus on observational epidemiological data, it’s important to align the text, figure, and simulations of Vignette 1 to properly reflect confounding among S_t , I_t , and β_t (or $C_{(t+1)}$) that would bias the effect estimators.
- b. It was unclear throughout most of the text whether the authors were referring to the observed incidence rate ($C_{(t+1)}^{(O)}$) or the unobserved incidence rate ($C_{(t+1)}$); a reader would have to infer the authors’ intent. Moreover, half of vignette 1 seems to have nothing to do with “measurement bias,” as emphasized in the vignette title, but instead focuses on correlations between transmission and incidence (lines 162-176). It is well-known that lack of correlation does not imply lack of causation, so this exercise does not appear to meaningfully link to the discussion of bias introduced by the vignette.

2. Vignette 2: This vignette is titled “Confounding bias…” but it does not illustrate how spatial heterogeneity may bias the effect estimators, but instead refers exclusively to correlation values (swapping from Spearman’s to Pearson’s correlation in the text for no clear reason), values that do not indicate anything about the causal effects, as is well-known. Moreover, an exercise in vignette 2 (lines 223-229) refers to making an incorrect assumption about the sources of spatial heterogeneity, but it fails to tell the reader that one may not know whether the assumption is correct in real observational data, nor does it tell the reader how causal frameworks may allow one to determine whether the assumption is plausible. Finally, the latter half of vignette 2 (lines 230-242) omits the environmental variables entirely from the model as an exercise, which is perplexing, given that the causal question posed throughout the manuscript is how best to estimate the effect of weather on infection. Maybe this should be a lesson in setting up and sticking to a single causal question when conducting a study? Otherwise, this exercise seems unconnected to a discussion about bias/confounding.

3. Writing:

- a. The introduction doesn’t do a great job of setting up the problem. Some of the sentences are cryptic and hard to parse even for someone who understands this literature, and others will be challenging to those unfamiliar with the topics (e.g., what does it mean to derive a statistical model from “theoretical causal reasoning”). Can you make the text more accessible to novices?
- b. Line 97: These two refs don’t focus on causal inference, do they? Wouldn’t Runge et al. (2019) in Nature Communications, or the more recent Runge et al. (2023) article in Nature Reviews Earth & Environment, be better citations for methods?
- c. Lines 181-183. Technically, bias is a property of an estimator (a design or estimation procedure) not an estimate. Thus, one cannot speak of “estimated causal effects” as being biased or unbiased. One can, however, write, as the authors do elsewhere in the manuscript, “estimate an effect with bias.”
- d. Lines 319-320: “a risk earlier described as…” – the Table 2 fallacy was never discussed in the main or supplementary text, so “earlier” is misleading.
- e. We recommend you make the following sentence a topic sentence rather than bury it in a paragraph: “Causal inference—the sub-field of statistics aiming at inferring causes from observational data—offers a principled approach to tackle these issues and strengthen evidence in observational research (26,27).” [perhaps elaborate on the “issues” to make the topic sentence clearer]
- f. It may be easier for a reader to understand what is coming in the manuscript if the authors used text from the Discussion that is easier to read (“Through a series of case studies, we illustrate how [concepts from the field of causal inference can] help assess study design (vignette 1), evaluate the risk of confounding bias (vignette 2), strategically choose a study’s location to achieve the set-up of a natural experiment (vignette 3), and interpret the direct and—sometimes paradoxical—indirect effects of meteorological variables on transmission (vignette 4).”

g. Vignette 2: Would it be easier for a reader if one started with a description of confounding and then gave, as an example, the description of spatial heterogeneities? What is the causal reasoning that's important in this vignette? Thinking about confounders, no? "More generally, this vignette demonstrates the need for explicit causal models to disentangle the mechanisms underlying spatial heterogeneities." How does it demonstrate this? What is an 'explicit causal model?'

h. Vignette 3: Even for someone who is very familiar with natural experiments and instrumental (surrogate) variables, this vignette is hard to read. We can't imagine a novice being able to read and understand it. It's unclear what contribution this vignette is making towards the better causal inferences touted in the introduction and discussion (except to point out that scholars can use quasi-experiments).

4. Target Audience Needs: Perhaps the target audience only includes the "data science" scholars and does not include the mathematical modelers or experimentalists, but finding a few ways to engage the latter two groups and help them see how causal inference frameworks can help their work or how the three "stools" of climate-disease research (mathematical modeling, experimental studies, observational studies) fit together would elevate this paper further (see for example, Schlueter et al. 2023, PNAS).

5. Model. We understand that the model is just an illustrative vehicle and not intended to capture all the complexities of the kinds of systems that climate-disease scholars may encounter, but we thought some issues should have been addressed, either in Supp Text or in the main text.

a. Many issues regarding observational epidemiologic data are ignored without clear indication to the reader, such as mismatch of observation intervals to the underlying dynamics (lines 130-132). Simplifying assumptions are fine for illustrating key points, but an author should indicate how those assumptions remove certain complications for making causal inferences that must be handled appropriately in real observational data.

b. Vignettes 1-4 do not use consistent R_0 and average immunity settings (particularly, settings for Vignette 2 are inconsistent with Vignette 3 and 4), and the logic for why the values change across vignettes is unclear. Since R_0 , μ , S_t , I_t , and β_t are all linked in the simulations (and thus affect any bias being highlighted by the vignettes), consistent values of R_0 would be more informative for the target audience.

6. Appropriateness for NE&E: We assume that the authors and editor determined that NE&E is a good fit for this manuscript. Yet given the target audience and the illustrative numerical example, we find it odd that this manuscript was submitted to NEE rather than a climate journal or a public health or infectious disease journal. When we use Google Search to identify articles that have been written on "infectious diseases" in NE&E ["infectious disease" source:Nature source:Ecology source:& source:Evolution], we see 82 articles but most focus on non-human diseases, zoonotic diseases or the huma-ecosystem (or biodiversity) interface, or they study disease through an evolutionary lens. Moreover, in the manuscript, the connection to ecology is mostly implicit and made explicit only at the very end ("Because phenology is a near-universal feature of life, such research may also lead to new insights into the ecology of infectious diseases.").

Reviewer #3 (Remarks to the Author):

Guevara et al explore how a causal inference framework can be used to elucidate potential issues in the study of climate and infectious disease. They present 4 specific examples where a causal framework could guide inference in this field.

I really enjoyed reading this manuscript. The topic is particularly timely given growing interest in research in this area. Vignette 1 elucidated clearly some of the issues with time series regressions of climate on incidence. Vignette 2 focused on possible misinterpretations of traveling waves versus climate effects. Vignette 3 focuses on how study location can impact parameter estimates due to climate variability. Vignette 4 focuses on disentangling the effects of correlated climate drivers using mediation analysis.

I have a few questions:

I tend to think of traditional "causal inference" approaches as primarily attempting to fit regression models e.g. in the field of biostats and econometrics. However, it seems here that the causal inference "lens" of the paper leads the authors to fit mechanistic disease models e.g. "For every replicate time series, we then fitted the misspecified model—with a deterministic transmission model and stochastic observation model". I was wondering if the authors could comment more broadly on how their causal inference framework supports/(or other wise) traditional mechanistic disease modeling approaches?

Are approaches using only regression models always biased (vignette 1)? If we somehow estimate transmission first, then run the regression, does that still lead to bias? I am not completely clear if that is the approach the authors are taking in Vignette 3 - perhaps these methods could better clarified.

For Vignette 2: can traveling waves be mistaken as climate drivers? How might we distinguish between the two in observations? e.g. if we see traveling waves originate from urban areas, presumably this a better pointer to a traveling wave versus a climate effect?

In general, I would have appreciated a few more pointers throughout the manuscript on how to overcome some of the issues raised. Perhaps this could be pursued in the discussion, or in a paragraph after each vignette.

How might things change in a continuous time world?

Version 1:

Decision Letter:

3rd September 2024

Dear Dr Domenech de Cellès,

Your manuscript entitled "How causal inference concepts can guide research into the effects of climate on infectious diseases" has now been seen by two of our original reviewers, whose comments are attached. Reviewer 1 was unavailable to re-review.

The reviewers find the paper to be improved in revision, but Reviewer 2 (together with their co-reviewer) still has concerns about various aspects of the presentation of the manuscript, particularly for the Introduction. We do expect that this paper will be publishable in NEE, but we will need to see your responses to these remaining concerns, and some additional effort to improve the accessibility of this paper to a broad readership, before we will be able to make a final decision regarding publication. We may not need to send the revised version back to this reviewer another time, but we will decide that after we receive your files with the revised version of the paper and documenting your responses to the reviewer.

We have also decided that we will need to publish this as an Analysis paper, rather than as a Perspective. Our general description of an Analysis is "new analysis of existing data or describes new data obtained in a comparative analysis that leads to novel and arresting conclusions of importance to a broad audience", whereas a Perspective is broadly considered a variation on a Review paper. We understand there are some elements of literature review to your paper, but we feel the main contributions are in the Vignettes, and that these are novel and exploratory enough to constitute primary analysis and thus a better fit to an original, primary research format.

Analysis papers fall in our primary research category, which means that they are eligible for Open Access publication should you choose this option (we can provide further details about this). It also means that there should be a Methods section as part of the main text of the manuscript, rather than in the Supplementary Information. You may wish to provide further details of the modelling in this Methods section than are currently presented; Methods are not word-limited.

We typically ask for Analysis papers to be a maximum of 3500 words for the main text (Introduction, Results and Discussion - not including methods), but we understand that it may not be possible to shorten to this length at this point. Please see if any shortenings can be made as you revise the paper, but given that we are asking for this reformatting at this stage, we don't expect it to be as short as we would otherwise ask for. I feel it may be possible to shorten the Introduction section, and I hope that this format change may actually help with revising the paper for readability along the lines suggested by Reviewer 1.

Not much more reformatting should be needed. I have linked here to two other Analysis articles as examples (please let me know if you do not have access and I can send pdfs)

<https://www.nature.com/articles/s41559-023-02144-3>

<https://www.nature.com/articles/s41559-023-02257-9>

Other points:

We suggest removing 'How' from the title

Please cite all 33 papers included in the literature review in the main citation list

Please revise to avoid phrases such as 'In this perspective', given the format change.

I hope this sounds reasonable, but of course if you have any questions please just email me directly.

* If you have not done so already please begin to revise your manuscript so that it conforms to our Analysis format instructions at <http://www.nature.com/natecolevol/info/final-submission>. Refer also to any guidelines provided in this letter.

Link Redacted

Nature Ecology & Evolution is committed to improving transparency in authorship. As part of our efforts in this direction, we are now requesting that all authors identified as 'corresponding author' on published papers create and link their Open Researcher and Contributor Identifier (ORCID) with their account on the Manuscript Tracking System (MTS), prior to acceptance. ORCID helps the scientific community achieve unambiguous attribution of all scholarly contributions. You can create and link your ORCID from the home page of the MTS by clicking on 'Modify my Springer Nature account'. For more information please visit please visit www.springernature.com/orcid.

[redacted]

Reviewer comments:

Reviewer #2 (Remarks to the Author):

1. We appreciate the authors' efforts in revising Vignette 2 (now Vignette 3) and the authors' efforts to clarify some of the text in their responses to our review. However, we do not feel that the readability of the introduction and Vignette 1 has been improved by the modest revisions. Given the difficulty we had in inferring the authors' motivations from the introduction and in understanding their intent in Vignette 1, we had expected the authors to make some revisions to improve the readability of these sections and their connections to the other sections.

2. To this end, we also wish the authors had made better use of their DAG (Figure 1) to illustrate the causal concerns brought up in all four vignettes, as they purport to do in lines 133-134. DAGs are not only a useful tool for illustrating the true underlying causal model, but also for highlighting sources of bias in causal analyses. The DAG in Fig. 1 appears to be referenced in Vignette 1, but not mentioned again until Vignette 4, where it is only used as a callback to Vignette 1. If the authors used the DAG to highlight the issues they discuss in the text, we think the text would be more digestible to readers unfamiliar with causal frameworks.

3. We appreciate the authors' explanation of Vignette 1 but their clarification of the intent of this vignette brings up a new issue. Measurement bias has been a significant and well-known concern for SIR models for a while, with extensive attention spent on addressing the concern (e.g., Johndrow et al. 2020; Osthus et al. 2017; van Smeden et al. 2020). While we agree that measurement bias can be a threat to causal studies, it is not a unique focus of the causal inference literature (just as estimating standard errors appropriately or addressing sample selection bias are important issues, but not uniquely causal inference concerns). The prominent placement of the vignette on measurement bias as the first of the four vignettes led us to expect the authors were going to address a much more prominent concern in causal inference frameworks within that vignette: confounding. The appropriateness of a vignette on measurement bias in a perspective purporting to explain the importance of causal inference frameworks for epidemiologic models to reader is questionable, and we suggest the authors move this to vignette to the Supplement or reorder the vignettes such that this vignette is last.

4. We appreciate the authors' revised second section that describes a causal inference framework for readers. However, there is a discrepancy between the authors' definition of causal frameworks and their use of the term in the text. We view causal inference frameworks not as specific perspectives on a particular research question but instead a set of conceptual procedures and tools by which causal research questions are generated and evaluated. Researchers often refer to two main frameworks - the potential outcomes framework (developed by Donald Rubin) and the structural framework (developed by Judea Pearl) - but there are others. This definition of "causal inference framework" has been well-established in the literature across fields (see Pearl 2008; Pearl 2010; Ding and Li 2018; Yao et al. 2021; Hünemann and Bareinboim 2023; Runge et al. 2023 for just a few examples). The definition provided in the second section aligns with this perspective, but the use of the term in the manuscript is not consistent with the perspective. For example, the title of section 2 is misleading - the causal inference framework is not "illustrative" - a causal inference framework is applied to an illustrative example. Along those lines, "...we formulate a causal inference framework..." (line 112) should instead be something like "...we illustrate the application of a causal inference framework...". There may be more phrases like this in other parts of the text that we missed, so we encourage the authors to ensure the way they refer to causal inference frameworks is both in line with well-established literature and consistent through the text.

References

Ding, P., & Li, F. (2018). Causal Inference: A Missing Data Perspective. *Statistical Science*, 33(2), 214–237. <https://www.jstor.org/stable/26770992>

Hünermund, P. and Bareinboim, E. (2023). Causal inference and data fusion in econometrics, *The Econometrics Journal*, utad008, <https://doi.org/10.1093/ectj/utad008>

Johndrow, J., Ball, P., Gargiulo, M., & Lum, K. (2020). Estimating the Number of SARS-CoV-2 Infections and the Impact of Mitigation Policies in the United States. *Harvard Data Science Review*, (Special Issue 1). <https://doi-org.proxy1.library.jhu.edu/10.1162/99608f92.7679a1ed>

Osthus, D., Hickmann, K. S., Caragea, P. C., Higdon, D., & Del Valle, S. Y. (2017). Forecasting seasonal influenza with a state-space SIR model. *The annals of applied statistics*, 11(1), 202–224. <https://doi.org/10.1214/16-AOAS1000>

Pearl, J. (2010). Causal Inference. *Proceedings of Workshop on Causality: Objectives and Assessment at NIPS 2008*, in *Proceedings of Machine Learning Research* 6:39-58 Available from <https://proceedings.mlr.press/v6/pearl10a.html>.

Pearl, J. (2010). THE FOUNDATIONS OF CAUSAL INFERENCE. *Sociological Methodology*, 40: 75-149. <https://doi.org/10.1111/j.1467-9531.2010.01228.x>

Runge, J., Gerhardus, A., Varando, G. et al. (2023). Causal inference for time series. *Nat Rev Earth Environ* 4, 487–505. <https://doi.org/10.1038/s43017-023-00431-y>

van Smeden, M., Lash, T.L., and Groenwold, R.H.H (2020) Reflection on modern methods: five myths about measurement error in epidemiological research, *International Journal of Epidemiology*, Volume 49, Issue 1, Pages 338–347, <https://doi.org/10.1093/ije/dyz251>

Yao, L., Chu, Z., Li, S., Li, Y., Gao, J., and Zhang, A. (2021). A Survey on Causal Inference. *ACM Trans. Knowl. Discov. Data* 15, 5, Article 74 (October 2021). <https://doi.org/10.1145/3444944>

Reviewer #2 (Remarks on code availability):

We did not try to run the code, but we did use it to better understand what the authors were doing throughout the vignettes. So in that way, the code was useful.

Reviewer #3 (Remarks to the Author):

The authors have addressed all my comments. I think this manuscript is an interesting and timely contribution to the literature on climate and disease.

*****END*****

Version 2:

Decision Letter:

8th October 2024

Dear Dr. Domenech de Cellès,

Thank you for submitting your revised manuscript "Causal inference concepts can guide research into the effects of climate on infectious diseases" (NATECOLEVOL-24020447B). Thank you for revising the paper into the Analysis format. We are now happy in principle to publish it in *Nature Ecology & Evolution*, pending minor revisions to comply with any remaining editorial and formatting guidelines.

Thank you again for your interest in *Nature Ecology & Evolution*. Please do not hesitate to contact me if you have any questions.

[redacted]

Version 3:

Decision Letter:

31st October 2024

Dear Dr Domenech de Cellès,

We are pleased to inform you that your Analysis entitled "Causal inference concepts can guide research into the effects of climate on infectious diseases", has now been accepted for publication in *Nature Ecology & Evolution*.

Over the next few weeks, your paper will be copyedited to ensure that it conforms to *Nature Ecology and Evolution* style. Once your paper is typeset, you will receive an email with a link to choose the appropriate publishing options for your paper and our Author Services team will be in touch regarding any additional information that may be required

Due to the importance of these deadlines, we ask you please us know now whether you will be difficult to contact over the next month. If this is the case, we ask you provide us with the contact information (email, phone and fax) of someone who will be able to check the proofs on your behalf, and who will be available to address any last-minute problems. Once your paper has been scheduled for online publication, the Nature press office will be in touch to confirm the details.

Acceptance of your manuscript is conditional on all authors' agreement with our publication policies (see www.nature.com/authors/policies/index.html). In particular your manuscript must not be published elsewhere and there must be no announcement of the work to any media outlet until the publication date (the day on which it is uploaded onto our web site).

Please note that *Nature Ecology & Evolution* is a Transformative Journal (TJ). Authors may publish their research with us through the traditional subscription access route or make their paper immediately open access through payment of an article-processing charge (APC). Authors will not be required to make a final decision about access to their article until it has been accepted. [Find out more about Transformative Journals](https://www.springernature.com/gp/open-research/transformative-journals)

Authors may need to take specific actions to achieve [compliance](https://www.springernature.com/gp/open-research/funding/policy-compliance-faqs) with funder and institutional open access mandates. If your research is supported by a funder that requires immediate open access (e.g. according to [Plan S principles](https://www.springernature.com/gp/open-research/plan-s-compliance)) then you should select the gold OA route, and we will direct you to the compliant route where possible. For authors selecting the subscription publication route, the journal's standard licensing terms will need to be accepted, including [those licensing terms](https://www.nature.com/nature-portfolio/editorial-policies/self-archiving-and-license-to-publish) will supersede any other terms that the author or any third party may assert apply to any version of the manuscript.

We welcome the submission of potential cover material (including a short caption of around 40 words) related to your manuscript; suggestions should be sent to *Nature Ecology & Evolution* as electronic files (the image should be 300 dpi at 210 x 297 mm in either TIFF or JPEG format). Please note that such pictures should be selected more for their aesthetic appeal than for their scientific content, and that colour images work better than black and white or grayscale images. Please do not try to design a cover with the *Nature Ecology & Evolution* logo etc., and please do not submit composites of images related to your work. I am sure you will understand that we cannot make any promise as to whether any of your suggestions might be selected for the cover of the journal.

You can generate the link yourself when you receive your article DOI by entering it here: <http://authors.springernature.com/share>

it's been a pleasure working with you on this paper and I look forward to seeing it published soon.

[redacted]

P.S. Click on the following link if you would like to recommend Nature Ecology & Evolution to your librarian
<http://www.nature.com/subscriptions/recommend.html#forms>

** Visit the Springer Nature Editorial and Publishing website at http://editorial-jobs.springernature.com?utm_source=ejp_NEcoE_email&utm_medium=ejp_NEcoE_email&utm_campaign=ejp_NEcoE for more information about our career opportunities. If you have any questions please click [here](mailto:editorial.publishing.jobs@springernature.com).

Open Access This Peer Review File is licensed under a Creative Commons Attribution 4.0 International License, which permits use, sharing, adaptation, distribution and reproduction in any medium or format, as long as you give appropriate credit to the original author(s) and the source, provide a link to the Creative Commons license, and indicate if changes were made. In cases where reviewers are anonymous, credit should be given to 'Anonymous Referee' and the source.

Reviewer #1

This is a terrific paper, and very important. The authors' approach of simulating and then analyzing epidemics to identify methodological pitfalls is elegant. I would point out that while they refer to the studies of interest as observational, most of these studies use ecological exposure data which creates pitfalls, but also confers some strengths, and indeed opens up a longstanding environmental epidemiology toolbox as a useful array of approaches that can be applied to these problems.

I have a few relatively minor comments which I hope are useful to the authors.

We thank the reviewer for their encouraging and useful comments.

1. I think DAGs are incredibly helpful in making models and assumptions explicit. To the issue of ecological exposures (ubiquitous exposures such as weather, air pollutants, UV radiation and so forth), the problem of ecological fallacy is well recognized. I think the fact that use of ecological exposures makes observational studies invulnerable to confounding by case characteristics (which a DAG makes clear...health status, age and sex can't change the weather) is less commonly recognized.

We agree with these comments. In the revised text (see our response to the editor), we streamlined the text to re-emphasize this first key point: explicit causal frameworks in the form of DAGs are needed for studying the effect of weather on infectious diseases.

2. This is an interesting moment to be writing about climate change and models, because of course (in the pages of Nature no less) it has recently been noted that climate models themselves are beginning to fail, presumably as a result of as-yet unrecognized positive feedback loops in climatic systems. This may warrant mention. See: <https://www.nature.com/articles/d41586-024-00816-z>.

The reviewer raises an important point, and we agree that reliable climate models are a prerequisite to predicting the impact of climate change on infectious diseases. However, we believe the discussion of these models is beyond the scope of our study, so we prefer not to comment on this in the main text.

3. Seasonality is of course a ubiquitous attribute of communicable diseases (and it would be nice to cite what I think is now a bit of a landmark work on this by Micaela Martinez: <https://journals.plos.org/plospathogens/article?id=10.1371/journal.ppat.1007327>). But an important fix to this, which doesn't seem to be mentioned in the MS, is simply using smoothers (whether fast Fourier terms or cubic splines, or other trend terms) to eliminate nonspecific seasonal oscillation that can be driven by behaviour, immune effects or environmental effects. That means that we are effectively regressing on the RESIDUALS of environmental exposures...and we get coefficients interpretable as effects that attend

deviation from expected seasonal oscillation. If we don't do this we get predictable "pirates prevented climate change" sorts of correlations...but I think the importance of separating smooth and rough components of time series using smothers is pretty well recognized.

We thank the reviewer for pointing us to Micaela Martinez's work, which we agree is a landmark paper we cited in the initial submission (Ref. 15, which we mentioned to discuss the "calendar of epidemics"). Regarding the use of smoothers to eliminate non-specific oscillations, we agree such techniques (*e.g.*, in the form of GAMs) represent a major advance compared to more standard regression models. In Vignette 1, we tested using such smoothers (cubic splines) to capture the unobserved variations in the variable $\log S_t I_t$, which causally affected our outcome variable (see Fig. 1). As shown in Fig. 2. However, this smooth term did not reduce the bias in the estimated effects of climate, presumably because the smoother did not appropriately capture the unobserved variations. Admittedly, given the scope of this perspective, we did not assess whether this result was systematic. In addition, as the reviewer correctly points out, smoothers do a good job in other fields of ecological research. Nevertheless, we believe that, at least in infectious disease ecology, transmission models are generally more appropriate as they mechanistically represent the dynamic of the unobserved process (especially the size of the susceptible population, which sensitively controls transmission dynamics). In the revised manuscript (see also our response to the editor above), we now highlight this point as the second key message of our perspective.

4. For climate change effects we are somewhat held back by the lack of a "control planet"...we only have one planet. This led scholars like Hsiang to use large scale, irregular oceanic phenomena like ENSO as "climate change-like shocks" to evaluate likely climate change effects. It seems to me that on a DAG ENSO might look and awful lot like a randomized exposure or instrumental variable...

The reviewer is exactly right, and we now mention these ENSO studies as examples of quasi-experiments in vignette 2 (end of first paragraph).

5. I think there are other fixes to some of the issues the authors flag...the issue of multi-causality in infectious disease research and environmental exposures is pretty well recognized (*e.g.*, with Lyme disease we can see precipitation effects at very long lags (mouse populations), and much shorter lags (human behaviour)). Here the use of the environmental epidemiology toolbox (*e.g.*, distributed nonlinear lag models) can be really useful. Similarly, I think the issues around "pseudo-diffusion" that the authors flag would be readily addressed using multilevel models that treat geographies as fixed or random effects.

We agree with the reviewer's comment about the lags. For simplicity, the causal effects in our model were all fixed at a lag of 1 week (because we assumed a fixed generation time of one week), but we acknowledged in the discussion the need for more realistic lag structures in real-world applications.

The reviewer raises an interesting point about using multilevel models to tackle the issue of pseudo-diffusion (vignette 2, now 3). In the revised manuscript, we now present a new analysis where we fitted a Gaussian process model (which can be viewed as an extension of mixed-effects/multilevel models in which the groups are continuous) to the simulated incidence data across Colombia while including time smoothers to capture variations in the unobserved variables. We find that such a model still results in bias, as it overestimates spatial coupling even when the climatic variables are included as covariates (see updated Figure 4). Again, given the scope of this perspective, we cannot claim this result is systematic. However, it is generally consistent with our second key point and suggests that, for infectious diseases, the latent process is difficult to capture even with advanced but purely statistical approaches.

Reviewer #2

Summary: The manuscript aims to raise awareness, and briefly introduce, some key ideas from the literature on causal inference. The target audience seems to be a mix of epidemiologists and climate scientists who are interested in the nexus between climate change and infectious disease. The authors assert that this audience is unaware of key ideas from the last twenty to thirty years from scholarship aimed at improving the quality of causal inferences from data (i.e., innovations aimed at strengthening causal claims from observational data). To illustrate a few of these ideas and highlight how understanding them can improve empirical inferences that scholars draw about the causal relationships between weather and disease, the authors use four simulated examples (“vignettes”).

Reviewer Background: This review reflects a co-review by a social scientist and a natural scientist, both of whom study causal inference and have written articles that seek to communicate about causal inference methods to scholarly and practitioner audiences who are unfamiliar with these methods. One of us also has research experience in infectious disease, but neither of us works at the nexus of climate and disease. Thus, our review focuses on the accuracy of the text and our impressions of how likely the target audience for this manuscript will understand the concepts being presented, based on our experience communicating with scientists unfamiliar with causal inference frameworks. Given our background, we are not able to determine whether the ideas described in this manuscript are already well-known to researchers working at the climate-disease nexus. Our review assumes that the authors are correct that most researchers working in this area are not familiar with these ideas despite the fact that there are articles about similar ideas targeted at climate scientists (e.g., Runge et al., 2019, Nat Comm) and epidemiologists (many articles, including some that include a SIR/SIRS model; eg., Ackley et al., 2017, Epi Methods).

We thank the reviewers for their thorough reading and detailed comments. In the revised manuscript, we now cite the important paper by Runge et al. Please note that we already cited work by Ackley et al. (Ref. 51 in the initial submission), but we also added the suggested reference to the revised manuscript. While we agree that some research in this direction has emerged over the past years, our general point is that such research has not yet become mainstream (as it should be, in our opinion). Our mini-review certainly supports this claim, as it shows that causal thinking is almost altogether absent from recent observational studies on climate and infectious diseases.

Main Comments

1. Errors: Apart from Vignette 1, we did not find any serious errors in the text.

Thank you for your comments on vignette 1—please see our detailed response below. However, we respectfully point out that while the underlying causal problem may have

different interpretations, this vignette does not contain any error, as can be checked from the codes provided in the submission.

2. Readability: Our main concern with the manuscript is whether the targeted audience will be able to understand the text. The intent of the manuscript is to illustrate the value of viewing research questions in this field through the lens of a “causal inference framework.” Yet, the text doesn’t clarify what a “causal inference framework” is and, in all four vignettes, does not explain how readers are supposed to use such a framework to improve their analysis. If we (the reviewers) hadn’t been so familiar with the ideas being presented, we doubt that we would have been able to follow the text. We expect that, to the novice reader, the vignettes will read like a set of objects that the authors have pulled from a larger bag of objects that is never described. The ideas in the vignettes are indeed conceptually connected under popular causal inference frameworks, but the manuscript doesn’t make these connections clear in the text. So, we worry that the thread of the argument may be difficult for a reader from the target audience to follow. Ultimately, the best review on readability will be from members of the target audience who can say, “Yes, this information is new to me or would be to my collaborators, and, yes, I can understand what the authors are saying.” But, if we’re correct about the readability challenge of this manuscript, the value of the manuscript drops considerably. We can see two potential solutions that may help a bit.

a. Completely revise to focus more on the stated aim: the value of causal inference frameworks (the constraints of the NE&E Perspective format may not be conducive to introducing these important ideas to a novice audience).

b. Reduce some of the text and focus on one or two of the subjects of the vignettes rather than four of them (like 1 and 2, which are connected). And then do a better job of explaining the topics and highlighting better in the intro that these topics are neglected in the literature and that publications targeted at epidemiologists and climate scientists on the utility of causal inference frameworks have not covered these topics in a way that is relevant for climate-disease scientists. To find more space for explanations, move some text to the Supplement: the text from lines 99-140, which explains details of the models and the justifications for its features, and the text from lines 333-356, which highlights limitations of the model and the vignettes (in main text, you can just say that real estimation is more challenging and that’s all the more reason to use a clear causal framework, and then provide more details in Supp). Use the saved text to do a better job of explaining the key ideas and their relevance to the target audience.

We thank our reviewers for this valuable comment, and with hindsight, we agree that the readability could have been improved. In response to the reviewers and following the editor’s recommendation to keep all the vignettes, we opted for the suggested solution a. Specifically, we completely rewrote the second section (formerly entitled “Illustrative infectious disease model”) to highlight the value of causal inference frameworks better. In this revised section (now entitled “Illustrative causal inference framework”), we now introduce and explicitly define causal inference frameworks and explain how we encapsulated such a framework into

our illustrative transmission model. We also better highlight our two key points: first, causal inference frameworks are required to assess the effects of weather on infectious diseases; second, transmission models are valuable to encapsulate this framework, as they specify explicit causal mechanisms for the observed and unobserved variables that underlie infectious disease data. We also revised the conclusion of every vignette to emphasize how the causal inference concept discussed illustrates these two points. Overall, we believe these changes have greatly helped tighten the thread of our perspective and make it accessible to a large audience not necessarily familiar with causal inference. As in the initial submission, we also use the glossary table to list and explicitly define the technical terms that may not be familiar to all readers and guide them through the text.

Other Comments

1. Vignette 1:

a. If the underlying (data-generation) procedure were exactly as shown in Figure 1 and described in Vignette 1, removing a term that includes S_t and/or I_t from the model should not result in biased effect estimators, since neither are shown or discussed as confounders in the figure or text. However, the transmission rate β_t is a function of the average transmission rate β (as given in the Supplementary Text), which itself is a function of R_0 and μ (according to the R code linked in the manuscript). The parameter μ influences the replenishment of the total population, thus influencing both S_t and I_t , which is likely where the confounding is being introduced. If this is indeed the case, none of these key details are reflected in Vignette 1 or Figure 1. Since transmission rates are usually estimated from prevalence and incidence data, it would be expected that S_t and I_t have some influence on β that is not reflected by Vignette 1. Given the manuscript's intent to focus on observational epidemiological data, it's important to align the text, figure, and simulations of Vignette 1 to properly reflect confounding among S_t , I_t , and β_t (or $C_{(t+1)}$) that would bias the effect estimators.

The reviewers raise an important point. While working on this vignette, we had extensive discussions with our co-author, Tobias Kurth, and other experts working on causal inference in Berlin. We concluded from these discussions that the causal problem was indeed measurement bias. Specifically, although the climatic variables causally affect the transmission rate, regression is performed on the incidence rate. Based on the DAG shown in Figure 1, the latter rate can be interpreted as a measurement of the former, with the two unobserved variables (numbers of susceptible and infected individuals) introducing non-random, systematic differences between the two rates—*i.e.*, measurement bias. To the best of our understanding, this causal problem does not result from confounding. Specifically, though the parameter μ indeed influences the replenishment of the population (via births), it does so for the S_t variable, but not the I_t variable (see the model equations). In epidemic models, the transmission rate depends on the intrinsic transmissibility of the pathogen and the contact patterns in the population. It is thus not directly affected by the variables S_t and I_t (see also the equation for β_t , which makes that point clear). In addition, what is usually estimated from incidence or prevalence data is not the transmission rate but the effective reproduction number. In the revised manuscript, we present a new analysis of time-series

regression for this outcome, in which we show that the problem of measurement bias, though less pronounced, persists in this case.

b. It was unclear throughout most of the text whether the authors were referring to the observed incidence rate ($C_{t+1}^{(O)}$) or the unobserved incidence rate (C_{t+1}); a reader would have to infer the authors' intent. Moreover, half of vignette 1 seems to have nothing to do with "measurement bias," as emphasized in the vignette title, but instead focuses on correlations between transmission and incidence (lines 162-176). It is well-known that lack of correlation does not imply lack of causation, so this exercise does not appear to meaningfully link to the discussion of bias introduced by the vignette.

Throughout the text, we used the terms "incidence rate" or "true incidence rate" for C_t and "observed incidence rate" for $C_t^{(O)}$. We have checked and revised the text to avoid any confusion. Regarding the correlations between the transmission and incidence rates, we did not report them to measure causality but to illustrate that these two rates may substantially differ, for example, because of non-linear effects resulting in sub-harmonic resonance. We believe this point is interesting and may not be familiar to all readers, so we prefer to keep this text as it is.

2. Vignette 2: This vignette is titled "Confounding bias..." but it does not illustrate how spatial heterogeneity may bias the effect estimators, but instead refers exclusively to correlation values (swapping from Spearman's to Pearson's correlation in the text for no clear reason), values that do not indicate anything about the causal effects, as is well-known. Moreover, an exercise in vignette 2 (lines 223-229) refers to making an incorrect assumption about the sources of spatial heterogeneity, but it fails to tell the reader that one may not know whether the assumption is correct in real observational data, nor does it tell the reader how causal frameworks may allow one to determine whether the assumption is plausible. Finally, the latter half of vignette 2 (lines 230-242) omits the environmental variables entirely from the model as an exercise, which is perplexing, given that the causal question posed throughout the manuscript is how best to estimate the effect of weather on infection. Maybe this should be a lesson in setting up and sticking to a single causal question when conducting a study. Otherwise, this exercise seems unconnected to a discussion about bias/confounding.

In response to the reviewers, we completely revised this vignette and conducted new analyses to illustrate the problem and propose a solution to the readers. Specifically, we no longer report correlation coefficients but instead the results of a Gaussian Process model fitted to all the data in Colombia. We find that this model, even when adding the climatic variables and a time smoother as covariates, overestimates the degree of spatial spread. We then discuss this result in connection with vignette 1 to highlight our second key point: purely statistical models may be limited in their ability to capture the latent processes (*e.g.*, variations in population immunity) that govern transmission dynamics. We then developed a new transmission model with climate forcing and spatial coupling between two locations to propose a solution (see Text S5). By fitting this model in several pairs of locations in

Colombia, we show how the spatial coupling parameter is correctly estimated to its true value (0), thus leading to the correct conclusion about the causal mechanisms underlying spatial heterogeneity in incidence.

3. Writing:

a. The introduction doesn't do a great job of setting up the problem. Some of the sentences are cryptic and hard to parse even for someone who understands this literature, and others will be challenging to those unfamiliar with the topics (e.g., what does it mean to derive a statistical model from "theoretical causal reasoning"). Can you make the text more accessible to novices?

We have revised the sentence containing "theoretical causal reasoning," which we agree was difficult to read. Without other specific examples, we find it hard to fully respond to the reviewers' comments. While writing the introduction, we deliberately made it as simple as possible to reach a wide audience and avoid unnecessary jargon. As a result, we believe the introduction is readable, but we would be happy to address more specific criticism. Please also note that, in response to your request, we wrote a new section to introduce the main concepts of causal inference.

b. Line 97: These two refs don't focus on causal inference, do they? Wouldn't Runge et al. (2019) in Nature Communications, or the more recent Runge et al. (2023) article in Nature Reviews Earth & Environment, be better citations for methods?

We have now added the two suggested references. We prefer to keep the other two references, as they are more specific to the research field of climate–infectious disease and cover some of the topics discussed in our perspective (though we agree they do not focus on causal inference *per se*).

c. Lines 181-183. Technically, bias is a property of an estimator (a design or estimation procedure) not an estimate. Thus, one cannot speak of "estimated causal effects" as being biased or unbiased. One can, however, write, as the authors do elsewhere in the manuscript, "estimate an effect with bias."

That is exactly right. We have corrected the text to avoid this potential confusion.

d. Lines 319-320: "a risk earlier described as..." – the Table 2 fallacy was never discussed in the main or supplementary text, so "earlier" is misleading.

We revised this sentence to clarify that we referred to earlier work by other research groups: "a risk described by earlier research as Table 2 fallacy [...]".

e. We recommend you make the following sentence a topic sentence rather than bury it in a paragraph: "Causal inference—the sub-field of statistics aiming at inferring causes from

observational data—offers a principled approach to tackle these issues and strengthen evidence in observational research (26,27).” [perhaps elaborate on the “issues” to make the topic sentence clearer]

In response (see also our response to your comment 2 above), we completely rewrote the second section to expand on these points.

f. It may be easier for a reader to understand what is coming in the manuscript if the authors used text from the Discussion that is easier to read (“Through a series of case studies, we illustrate how [concepts from the field of causal inference can] help assess study design (vignette 1), evaluate the risk of confounding bias (vignette 2), strategically choose a study’s location to achieve the set-up of a natural experiment (vignette 3), and interpret the direct and—sometimes paradoxical—indirect effects of meteorological variables on transmission (vignette 4).”

At the end of the new section entitled “Illustrative causal inference framework”, we now introduce the four causal inference concepts covered and our two key points regarding the need for explicit causal frameworks and transmission models.

g. Vignette 2: Would it be easier for a reader if one started with a description of confounding and then gave, as an example, the description of spatial heterogeneities? What is the causal reasoning that’s important in this vignette? Thinking about confounders, no? “More generally, this vignette demonstrates the need for explicit causal models to disentangle the mechanisms underlying spatial heterogeneities.” How does it demonstrate this? What is an ‘explicit causal model?’

In every vignette, we start with a non-technical description of the causal problem, then name it based on causal inference theory, and finally illustrate it with our transmission model. We thus prefer to keep this text as it is to maintain the logical structure of the text. Regarding the meaning of “explicit causal models,” please note that we ran new analyses and extensively revised this vignette (see our response to the editor above). In particular, we present results from a new transmission model with climate forcing and spatial coupling between two locations—such a model illustrates what we meant by explicit causal models.

h. Vignette 3: Even for someone who is very familiar with natural experiments and instrumental (surrogate) variables, this vignette is hard to read. We can’t imagine a novice being able to read and understand it. It’s unclear what contribution this vignette is making towards the better causal inferences touted in the introduction and discussion (except to point out that scholars can use quasi-experiments).

Please note that we did not discuss instrumental variables in this vignette. Our general point is that different climates have different properties, which may usefully be viewed as natural experiments to isolate the individual effects of weather variables. We then illustrate this point by comparing a tropical climate and a temperate climate, showing that the former makes it

easier to identify the effect of relative humidity and the latter the effect of temperature on transmission. Because of the extensive debates about the relative roles of temperature and humidity (for example, on influenza seasonal epidemiology), we believe this vignette may be highly valuable to the field and help researchers make strategic choices about their study's location, depending on their causal question of interest. Regarding the reviewers' comment that the text is hard to read, we have tried to simplify it whenever possible.

4. Target Audience Needs: Perhaps the target audience only includes the "data science" scholars and does not include the mathematical modelers or experimentalists, but finding a few ways to engage the latter two groups and help them see how causal inference frameworks can help their work or how the three "stools" of climate-disease research (mathematical modeling, experimental studies, observational studies) fit together would elevate this paper further (see for example, Schlueter et al. 2023, PNAS).

As explained above, we have revised the text to emphasize our second key point: mathematical transmission models are valuable in encapsulating causal inference frameworks, as they specify explicit mechanisms for the observed and unobserved variables that govern infection, transmission, and disease dynamics. This point, in addition to the illustrative transmission model we use throughout the paper, should fully engage infectious disease modelers. Regarding the experimentalists, we indeed focus on research based on observational data. However, we believe this is not a major limitation, as most data generated in infectious disease ecology and epidemiology are observational.

5. Model. We understand that the model is just an illustrative vehicle and not intended to capture all the complexities of the kinds of systems that climate-disease scholars may encounter, but we thought some issues should have been addressed, either in Supp Text or in the main text.

The reviewers are exactly right: We designed a deliberately simple transmission model to illustrate the different causal concepts. In doing so, we ignored many practical complexities that inevitably arise during real-world data analysis. Please note that we explicitly acknowledged this limitation and listed the different methodological complexities in the discussion, in the paragraph starting with the sentence: "Because of this perspective's conceptual focus, we sidestepped the many methodological technicalities that inevitably arise in practice."

a. Many issues regarding observational epidemiologic data are ignored without clear indication to the reader, such as mismatch of observation intervals to the underlying dynamics (lines 130-132). Simplifying assumptions are fine for illustrating key points, but an author should indicate how those assumptions remove certain complications for making causal inferences that must be handled appropriately in real observational data.

We added the reviewers' comment about the mismatch between observation intervals and infection dynamics in the paragraph discussed above.

b. Vignettes 1-4 do not use consistent R_0 and average immunity settings (particularly, settings for Vignette 2 are inconsistent with Vignette 3 and 4), and the logic for why the values change across vignettes is unclear. Since R_0 , μ , S_t , I_t , and β_t are all linked in the simulations (and thus affect any bias being highlighted by the vignettes), consistent values of R_0 would be more informative for the target audience.

Although most parameters were fixed across vignettes, we indeed chose to vary a few parameters to get simulations that produced more interesting/relevant dynamics for the vignette considered. As we wrote in the model's description, we do not focus on a particular pathogen, so we believe this choice is justified. In addition, we carefully listed all the parameter values in Table S3 and each figure's legend to avoid any confusion for the reader. Finally, the focus of our perspective is conceptual, so we believe that exact parameter values are less important than the causal concepts we discuss.

6. Appropriateness for NE&E: We assume that the authors and editor determined that NE&E is a good fit for this manuscript. Yet given the target audience and the illustrative numerical example, we find it odd that this manuscript was submitted to NEE rather than a climate journal or a public health or infectious disease journal. When we use Google Search to identify articles that have been written on “infectious diseases” in NE&E [“infectious disease” source:Nature source:Ecology source:& source:Evolution], we see 82 articles but most focus on non-human diseases, zoonotic diseases or the huma-ecosystem (or biodiversity) interface, or they study disease through an evolutionary lens. Moreover, in the manuscript, the connection to ecology is mostly implicit and made explicit only at the very end (“Because phenology is a near-universal feature of life, such research may also lead to new insights into the ecology of infectious diseases.”).

We appreciate the reviewers' comments, but we point out that neither the editor nor the other two reviewers—both apparently experts in infectious disease ecology—raised a similar concern. As we wrote in the introduction, the climate is a ubiquitous environmental exposure for all living organisms and, thus, a central topic in ecology. Hence, we believe our perspective is perfectly within the scope of *Nature Ecology & Evolution*.

Reviewer #3

Guevara et al explore how a causal inference framework can be used to elucidate potential issues in the study of climate and infectious disease. They present 4 specific examples where a causal framework could guide inference in this field.

I really enjoyed reading this manuscript. The topic is particularly timely given growing interest in research in this area. Vignette 1 elucidated clearly some of the issues with time series regressions of climate on incidence. Vignette 2 focused on possible misinterpretations of traveling waves versus climate effects. Vignette 3 focuses on how study location can impact parameter estimates due to climate variability. Vignette 4 focuses on disentangling the effects of correlated climate drivers using mediation analysis.

I have a few questions:

I tend to think of traditional "causal inference" approaches as primarily attempting to fit regression models e.g. in the field of biostats and econometrics . However, it seems here that the causal inference "lens" of the paper leads the authors to fit mechanistic disease models e.g. "For every replicate time series, we then fitted the misspecified model—with a deterministic transmission model and stochastic observation model". I was wondering if the authors could comment more broadly on how their causal inference framework supports/(or other wise) traditional mechanistic disease modeling approaches?

The reviewer is exactly right: though causal inference is more often used in biostatistics and econometrics, our perspective aims to provide an effort to apply these tools for climate–infectious disease research. Traditional mechanistic modeling approaches are also causal tools but are not typically framed into a causal framework. For infectious disease modelers, we believe the concepts covered in vignettes 2 (natural experiments) and 4 (direct and indirect effects of temperature) will be especially new and useful. In the revised manuscript, we now elaborate on these points in the new section entitled “Illustrative causal inference framework.”

Are approaches using only regression models always biased (vignette 1)? If we somehow estimate transmission first, then run the regression, does that still lead to bias? I am not completely clear if that is the approach the authors are taking in Vignette 3 - perhaps these methods could better clarified.

The reviewer raises an important point regarding vignette 1. Although we focused on incidence rates as outcomes, an alternative and common study design is to first reconstruct the effective reproduction numbers (R_e) from the observed incidence rates and then regress the R_e estimates against weather variables. In response to the reviewer, we ran new analyses and tested time series regression (TSR) for this outcome. As shown in the new Figure S2, the DAG corresponding to R_e illustrates that this outcome variable only depends on one

unobserved variable, while the incidence rate depends on two. Accordingly, one would expect TSR to perform better for this outcome. Rerunning our simulation study, we find this is indeed the case, though the effects of temperature and relative humidity are still estimated with a large bias. Admittedly, given the scope of this perspective, we did not assess whether this result was systematic, but these results suggest that TSR based on R_e is indeed a better strategy. We now report these new results in the revised manuscript (new paragraph in vignette 1, updated figure S1, and new figure S2).

For Vignette 2: can traveling waves be mistaken as climate drivers? How might we distinguish between the two in observations? e.g. if we see traveling waves originate from urban areas, presumably this a better pointer to a traveling wave versus a climate effect?

We appreciate the reviewer's insightful comments. To investigate if spatial diffusion can bias the effect of weather on transmission, we conducted simulations using our two-location transmission model (new Text S5). For the simulations, we incorporated weather conditions specific to each location ($\delta_{Te} = \delta_{RH} = -0.2$) while varying the spatial diffusion parameter ($\tau = 0-1$). We then estimated the effect of weather by backfitting our model while assuming no spatial diffusion ($\tau = 0$). Our findings revealed that neglecting spatial diffusion leads to an underestimation of the effect of weather on transmission (Likelihood profiles for relative humidity in Figure R1 below).

However, we think that the causal mechanism is different from that described in vignette 2 (now 3), where weather confounds the estimation of spatial diffusion (measured as the effect of $C_{t+1}^{(0,1)} \rightarrow C_{t+1}^{(0,2)}$) because it is a shared cause in both locations ($C_{t+1}^{(0,1)} \leftarrow RH \rightarrow C_{t+1}^{(0,2)}$). Here, we change the causal question as we are interested in estimating the effect of climate on the transmission rate (measured as the effect of $RH \rightarrow C_{t+1}^{(0,1)}$). Since spatial diffusion does not affect relative humidity, it cannot be a confounder. Instead, we think it introduces a new causal path from climate to incidence ($RH \rightarrow C_{t+1}^{(0,2)} \rightarrow C_{t+1}^{(0,1)}$). Thus, the total effect of climate on transmission in one location is a combination of its direct effect in the location ($RH \rightarrow C_{t+1}^{(0,1)}$) and an indirect effect mediated through the second location ($RH \rightarrow C_{t+1}^{(0,2)} \rightarrow C_{t+1}^{(0,1)}$). While our findings provide valuable insights, we think that a more comprehensive exploration of these results, especially in the context of urban versus rural dynamics where population sizes and interactions differ, is warranted. We have highlighted these limitations and future research directions in the discussion of our vignette.

Figure R1. Spatial diffusion between Bogotá and Pereira, Colombia, can bias the estimation of the effect of climate on transmission. We simulated using transmission models incorporating the effect of spatial diffusion ($\tau = 0-1$) and fitted transmission models neglecting the effect of spatial diffusion. The vertical grey line is the true value of the direct effect of relative humidity ($\delta_{RH} = -0.2$), and the right values indicate the fixed values of τ .

In general, I would have appreciated a few more pointers throughout the manuscript on how to overcome some of the issues raised. Perhaps this could be pursued in the discussion or in a paragraph after each vignette.

We thank the reviewer for this comment, which was also raised by reviewer 2. In response, we revised the conclusions of every vignette to clarify how to overcome the issues raised. In so doing, we also emphasize our second key point about the value of transmission models for encapsulating the causal inference frameworks we advocate.

Reviewer #2

1. We appreciate the authors' efforts in revising Vignette 2 (now Vignette 3) and the authors' efforts to clarify some of the text in their responses to our review. However, we do not feel that the readability of the introduction and Vignette 1 has been improved by the modest revisions. Given the difficulty we had in inferring the authors' motivations from the introduction and in understanding their intent in Vignette 1, we had expected the authors to make some revisions to improve the readability of these sections and their connections to the other sections.

Following correspondence with the editor to ask how best to address this comment, Dr. Turner suggested edits to simplify and clarify the introduction. In particular, following Dr. Turner's recommendation, we divided the introduction into sub-sections with heading titles to guide the readers and facilitate the reading. We hope these changes will satisfy the reviewers.

2. To this end, we also wish the authors had made better use of their DAG (Figure 1) to illustrate the causal concerns brought up in all four vignettes, as they purport to do in lines 133-134. DAGs are not only a useful tool for illustrating the true underlying causal model, but also for highlighting sources of bias in causal analyses. The DAG in Fig. 1 appears to be referenced in Vignette 1, but not mentioned again until Vignette 4, where it is only used as a callback to Vignette 1. If the authors used the DAG to highlight the issues they discuss in the text, we think the text would be more digestible to readers unfamiliar with causal frameworks.

We agree that DAGs are useful for highlighting sources of bias—we now explicitly make that point in the revised "Causal inference framework" section. In response, we also separated Table 1 into two tables, including one (new Table 1) focusing on causal inference concepts. In this new table, we added the DAG representing the causal concept discussed in every vignette. As the reviewers correctly point out, we believe this companion table will help highlight the issues addressed in the text and make the text more accessible to readers unfamiliar with causal inference.

We also double-checked all our references to DAGs, which did appear in vignettes 1, 3, and 4:

- Vignette 1, in which we cited figures 1 and S2 to illustrate the problem of measurement bias for two different endpoints, incidence rates (figure 1) and effective reproduction number (figure S2).
- Vignette 3, in which we cited figure S3 to illustrate the problem of confounding (we now present a simplified version of figure S3 in Table 1).
- Vignette 4, in which we cited Figure 1 to discuss mediators and direct and indirect effects.

That said, the reviewers correctly pointed out that vignette 2 lacked a reference to DAGs—this has been corrected in the revised manuscript (see also Table 1, where we added a DAG to describe quasi-experiments, the causal concept discussed in this vignette).

3. We appreciate the authors' explanation of Vignette 1 but their clarification of the intent of this vignette brings up a new issue. Measurement bias has been a significant and well-known concern for SIR models for a while, with extensive attention spent on addressing the concern (e.g., Johndrow et al. 2020; Osthus et al. 2017; van Smeden et al. 2020). While we agree that measurement bias can be a threat to causal studies, it is not a unique focus of the causal inference literature (just as estimating

standard errors appropriately or addressing sample selection bias are important issues, but not uniquely causal inference concerns). The prominent placement of the vignette on measurement bias as the first of the four vignettes led us to expect the authors were going to address a much more prominent concern in causal inference frameworks within that vignette: confounding. The appropriateness of a vignette on measurement bias in a perspective purporting to explain the importance of causal inference frameworks for epidemiologic models to reader is questionable, and we suggest the authors move this to vignette to the Supplement or reorder the vignettes such that this vignette is last.

We thank the reviewers for pointing us to these references, which we now cite in the main text. As far as we understand, these papers discuss reporting bias, that is, sources of bias that may introduce systematic differences between the true incidence rate C_t and the observed incidence rate $C_t^{(O)}$. In our simple illustrative model, we consider under-reporting but with no reporting bias, as we assume our observation model is unbiased ($C_t^{(O)}$ is simply a noisy, scaled-down version of C_t). We now mention this point while describing the observation model.

We believe the issue of measurement bias—*i.e.*, systematic differences between the targeted endpoint β_t (transmission rate) and the observed endpoint $C_t^{(O)}$ (observed incidence rate)—may be far more severe than reporting bias. Our mini-review also suggests this problem is less appreciated than reporting bias, which may explain the persisting popularity of time-series regression study designs. Regarding the position of this vignette, we put it first to illustrate the shortcomings of purely statistical models, a point we then repeat in the other vignettes where we use mechanistic models. Hence, the vignettes follow a logical order, and we do not aim to claim from the position of vignette 1 that measurement bias is the most important causal concept (compared with, for example, confounding bias). Therefore, we prefer to keep this vignette in its current position in the main text, a choice we validated with the editor.

4. We appreciate the authors' revised second section that describes a causal inference framework for readers. However, there is a discrepancy between the authors' definition of causal frameworks and their use of the term in the text. We view causal inference frameworks not as specific perspectives on a particular research question but instead a set of conceptual procedures and tools by which causal research questions are generated and evaluated. Researchers often refer to two main frameworks - the potential outcomes framework (developed by Donald Rubin) and the structural framework (developed by Judea Pearl) - but there are others. This definition of "causal inference framework" has been well-established in the literature across fields (see Pearl 2008; Pearl 2010; Ding and Li 2018; Yao et al. 2021; Hünermund and Bareinboim 2023; Runge et al. 2023 for just a few examples). The definition provided in the second section aligns with this perspective, but the use of the term in the manuscript is not consistent with the perspective. For example, the title of section 2 is misleading – the causal inference framework is not "illustrative" – a causal inference framework is applied to an illustrative example. Along those lines, "...we formulate a causal inference framework..." (line 112) should instead be something like "...we illustrate the application of a causal inference framework...". There may be more phrases like this in other parts of the text that we missed, so we encourage the authors to ensure the way they refer to causal inference frameworks is both in line with well-established literature and consistent through the text.

We agree with this comment. In response and to clarify our terminology, we revised the text as follows:

- We added the reference proposed (Pearl, 2010) to support our definition of causal inference frameworks in the second section.

- We revised the title of this section to: “Causal inference framework: Illustration and application to a transmission model of infectious diseases.”
- We revised line 112 and double-checked the text along the lines suggested by the reviewers to avoid potential confusion about the meaning of causal inference frameworks.

References

- Ding, P., & Li, F. (2018). Causal Inference: A Missing Data Perspective. *Statistical Science*, 33(2), 214–237. <https://www.jstor.org/stable/26770992>
- Hünernmund, P. and Bareinboim, E. (2023). Causal inference and data fusion in econometrics, *The Econometrics Journal*, utad008, <https://doi.org/10.1093/ectj/utad008>
- Johndrow, J., Ball, P., Gargiulo, M., & Lum, K. (2020). Estimating the Number of SARS-CoV-2 Infections and the Impact of Mitigation Policies in the United States. *Harvard Data Science Review*, (Special Issue 1). <https://doi-org.proxy1.library.jhu.edu/10.1162/99608f92.7679a1ed>
- Osthus, D., Hickmann, K. S., Caragea, P. C., Higdon, D., & Del Valle, S. Y. (2017). Forecasting seasonal influenza with a state-space SIR model. *The annals of applied statistics*, 11(1), 202–224. <https://doi.org/10.1214/16-AOAS1000>
- Pearl, J. (2010). Causal Inference. Proceedings of Workshop on Causality: Objectives and Assessment at NIPS 2008, in *Proceedings of Machine Learning Research* 6:39-58 Available from <https://proceedings.mlr.press/v6/pearl10a.html>.
- Pearl, J. (2010). THE FOUNDATIONS OF CAUSAL INFERENCE. *Sociological Methodology*, 40: 75-149. <https://doi.org/10.1111/j.1467-9531.2010.01228.x>
- Runge, J., Gerhardus, A., Varando, G. et al. (2023). Causal inference for time series. *Nat Rev Earth Environ* 4, 487–505. <https://doi.org/10.1038/s43017-023-00431-y>
- van Smeden, M., Lash, T.L., and Groenwold, R.H.H (2020) Reflection on modern methods: five myths about measurement error in epidemiological research, *International Journal of Epidemiology*, Volume 49, Issue 1, Pages 338–347, <https://doi.org/10.1093/ije/dyz251>
- Yao, L., Chu, Z., Li, S., Li, Y., Gao, J., and Zhang, A. (2021). A Survey on Causal Inference. *ACM Trans. Knowl. Discov. Data* 15, 5, Article 74 (October 2021). <https://doi.org/10.1145/3444944>

Reviewer #2 (Remarks on code availability):

We did not try to run the code, but we did use it to better understand what the authors were doing throughout the vignettes. So in that way, the code was useful.

Thank you. Please note that, should the paper be accepted, we will make all the codes available on GitHub and Edmond, the Open Data Repository of the Max Planck Society.

Reviewer #3

The authors have addressed all my comments. I think this manuscript is an interesting and timely contribution to the literature on climate and disease.

Thank you.